# A potent and selective reaction hijacking inhibitor of *Plasmodium falciparum* tyrosine tRNA synthetase exhibits single dose oral efficacy *in vivo*

Stanley C. Xie[1,2☯], Chia-Wei Tai[1☯], Craig J. Morton[3], Liting Ma[4], Shih-Chung Huang[4], Sergio Wittlin[5,6], Yawei Du[1], Yongbo Hu[4], Con Dogovski[1], Mina Salimimarand[7], Robert Griffin[4], Dylan England[4], Elisa de la Cruz[4], Ioanna Deni[8,9], Tomas Yeo[8,9], Anna Y. Burkhard[8,9], Josefine Striepen[8,9], Kyra A. Schindler[8,9], Benigno Crespo[10], Francisco J. Gamo[10], Yogesh Khandokar[11], Craig A. Hutton[7], Tayla Rabie[12], Lyn-Marié Birkholtz[12], Mufuliat T. Famodimu[13], Michael J. Delves[13], Judith Bolsher[14], Karin M. J. Koolen[14], Rianne van der Laak[14], Anna C. C. Aguiar[15], Dhelio B. Pereira[16], Rafael V. C. Guido[17], Darren J. Creek[2], David A. Fidock[8,9,18], Lawrence R. Dick[19], Stephen L. Brand[20], Alexandra E. Gould[4,21], Steven Langston[4‡*], Michael D. W. Griffin[1‡*], Leann Tilley[1*]

1 Department of Biochemistry and Pharmacology, Bio21 Molecular Science and Biotechnology Institute, The University of Melbourne, Melbourne, Victoria, Australia, 2 Drug Delivery Disposition and Dynamics, Monash Institute of Pharmaceutical Sciences, Monash University, Parkville, Victoria, Australia, 3 Biomedical Manufacturing Program, CSIRO, Clayton South, Victoria, Australia, 4 Takeda Development Center Americas, Inc., Cambridge, Massachusetts, United States of America, 5 Swiss Tropical and Public Health Institute, Allschwil, Switzerland, 6 University of Basel, Basel, Switzerland, 7 School of Chemistry, The University of Melbourne, Melbourne, Victoria, Australia, 8 Center for Malaria Therapeutics and Antimicrobial Resistance, Columbia University Medical Center, New York, New York, United States of America, 9 Department of Microbiology and Immunology, Columbia University Medical Center, New York, New York, United States of America, 10 Global Health Medicines R&D, GSK, Tres Cantos, Madrid, Spain, 11 ANSTO-Australian Synchrotron, Clayton, Victoria, Australia, 12 Department of Biochemistry, Genetics and Microbiology, University of Pretoria Institute for Sustainable Malaria Control, University of Pretoria, Hatfield, South Africa, 13 Department of Infection Biology, London School of Hygiene and Tropical Medicine, London, United Kingdom, 14 TropIQ Health Sciences, Nijmegen, The Netherlands, 15 Federal University of São Paulo, São Paulo, São Paulo, Brazil, 16 Research Center for Tropical Medicine of Rondonia, Porto Velho, Brazil, 17 Sao Carlos Institute of Physics, University of São Paulo, São Carlos, Brazil, 18 Division of Infectious Diseases, Department of Medicine, Columbia University Medical Center, New York, New York, United States of America, 19 Seofon Consulting, Natick, Massachusetts, United States of America, 20 Medicines for Malaria Venture, Geneva, Switzerland, 21 Broad Institute of MIT and Harvard, Cambridge, Massachusetts, United States of America

☯ These authors contributed equally to this work.
‡ These authors jointly supervised this work.
* Steve.Langston@takeda.com (SL); mgriffin@unimelb.edu.au (MDWG); ltilley@unimelb.edu.au (LT)

**Data Availability Statement:** The following structures have been deposited in the PDB: PfTyrRS/Tyr-ML471 - PDB ID 9CLL (https://doi.

## Abstract

The *Plasmodium falciparum* cytoplasmic tyrosine tRNA synthetase (*Pf*TyrRS) is an attractive drug target that is susceptible to reaction-hijacking by AMP-mimicking nucleoside sulfamates. We previously identified an exemplar pyrazolopyrimidine ribose sulfamate, ML901, as a potent reaction hijacking inhibitor of *Pf*TyrRS. Here we examined the stage specificity of action of ML901, showing very good activity against the schizont stage, but lower trophozoite stage activity. We explored a series of ML901 analogues and identified ML471, which exhibits improved potency against trophozoites and enhanced selectivity against a human

org/10.2210/pdb9CLL/pdb).Additional data and source data are available in Supplementary Information.

**Funding:** Funding was provided by the Global Health Innovative Technology Fund, Japan (H2019-104 to LT, LRD, SL, AEG), the Australian National Health and Medical Research Council (APP2022075 to LT), the Australian Research Council (DE230101173 to SCX), the Medicines for Malaria Venture (RD-19-001 to LMB; RD-08-0015 to DAF; RD-21-1003 to MJD), the Foundation for Research Support of the State of São Paulo (FAPESP; 2019/19708-0 and 2013/07600-3 to RVCG and ACCA), the South African Medical Research Council and the Department of Science and Innovation South African Research Chairs Initiative Grant managed by the National Research Foundation (UID 84627 to LMB), a Medical Research Council Career Development Award (MR/V010034/1 to MJD), the Bill & Melinda Gates Foundation (INV-033538 to DAF), and Millennium Pharmaceuticals, a wholly owned subsidiary of Takeda Pharmaceuticals Company Limited (LRD, SL, AEG). The funders had no role in study design, data collection and analysis, decision to publish, or preparation of the manuscript.

**Competing interests:** LM, SCH, YH, RG, DE, EdIC, SL are employees of Takeda Pharmaceuticals and owners of Takeda stock. The other authors have no competing interests to declare.

cell line. Additionally, it has no inhibitory activity against human ubiquitin-activating enzyme (UAE) *in vitro*. ML471 exhibits low nanomolar activity against asexual blood stage *P. falciparum* and potent activity against liver stage parasites, gametocytes and transmissible gametes. It is fast-acting and exhibits a long *in vivo* half-life. ML471 is well-tolerated and shows single dose oral efficacy in the SCID mouse model of *P. falciparum* malaria. We confirm that ML471 is a reaction hijacking inhibitor that is converted into a tight binding Tyr-ML471 conjugate by the *Pf*TyrRS enzyme. A crystal structure of the *Pf*TyrRS/ Tyr-ML471 complex offers insights into improved potency, while molecular docking into UAE provides a rationale for improved selectivity.

## Author summary

Malaria is a devastating disease caused by the *Plasmodium* parasite, with more than 200 million cases and 600,000 deaths reported in 2022. Worryingly, the emergence of clinically artemisinin-resistant *P. falciparum* was first identified in Southeast Asia and more recently detected in Africa. This imposes a significant burden on the healthcare systems of the world's poorest countries. This study continues our previous effort to develop a series of nucleoside sulfamates that targets the *P. falciparum* cytoplasmic tyrosine tRNA synthetase (*Pf*TyrRS), via a reaction hijacking mechanism. We explored the potency and selectivity of nucleoside sulfamate derivatives and identified a front runner compound, ML471, that shows potent activity against malaria parasites but low toxicity to human cells. Moreover, ML471 shows activity against multiple life stages of the parasite and exhibits single-dose oral efficacy in the SCID mouse model of *P. falciparum* malaria. We confirmed ML471 is a potent reaction hijacking inhibitor that is converted to Tyr-ML471 by the recombinant *Pf*TyrRS enzyme. The crystal structure of Tyr-ML471 bound *Pf*TyrRS offers molecular insights that could guide future drug design. Taken together, our findings provide a promising direction for developing new antimalarials.

## Introduction

Malaria is a debilitating disease caused by protist parasites of the genus *Plasmodium* that places an enormous health burden on the world's poorest communities. In 2022, more than 200 million people were infected with *P. falciparum*, resulting in more than 600,000 deaths [1]. The burden was exacerbated by disruptions to services during the COVID pandemic [2]. Unfortunately, the past 15 years have seen the emergence of *P. falciparum* parasites that exhibit partial resistance to artemisinin and partner drugs, such as piperaquine and mefloquine, resulting in ~50% treatment failure with standard artemisinin combination therapies in some regions of Southeast Asia [3,4]. The recent emergence in Africa of parasites harbouring artemisinin resistance-conferring K13 mutations [5–7] is of great concern, and the Medicines for Malaria Venture (MMV) not-for-profit partnership has declared that new antimalarial therapies and prophylaxis regimens need to be developed as a failsafe [8].

Certain AMP-mimicking nucleoside sulfamates act as reaction hijacking inhibitors of E1 enzymes, *i.e.* ubiquitin/ ubiquitin-like protein (UBL)-activating enzymes [9–13]. The UBL-bound form of these Adenylate-Forming Enzymes (AFEs) is susceptible to attack, leading to the formation of an inhibitory sulfamate-UBL adduct within the active site. This unusual

reaction hijacking mechanism has been exploited to generate new clinical candidates, such as Pevonedistat [14,15].

We previously screened a Takeda Pharmaceuticals nucleoside sulfamates library (Cambridge, MA, USA) and identified ML901 as an exemplar pyrazolopyrimidine sulfamate, with potent activity against *P. falciparum* [16]. Our group showed that, surprisingly, this AMP-mimicking nucleoside sulfamate uses a related reaction hijacking mechanism to target another AFE subclass, the aminoacyl tRNA synthetases. ML901 binds the *P. falciparum* cytoplasmic tyrosine tRNA synthetase (*Pf*TyrRS), and then reacts with the bound activated amino acid, resulting in the synthesis of an inhibitory sulfamate-amino acid adduct within the active site of the enzyme. By contrast, the equivalent human enzyme is not susceptible to reaction hijacking. That finding was the first demonstration of reaction hijacking of an enzyme class other than the E1 enzymes. More recently, we have identified an aminothienopyrimidine sulfonamide, OSM-S-106, with selective reaction hijacking activity against the *P. falciparum* asparagine tRNA synthetase [17].

ML901 exhibits low, but measurable toxicity against a mammalian cell line [16], which is potentially due to cross-inhibition of UBLs [12]. Thus, we explored a range of ML901 derivatives from the Takeda nucleoside sulfamate library, with different substitutions at the 7-position of the pyrazolopyrimidine ring system, to identify compounds with improved selectivity. We identified ML471 as a compound with enhanced potency against *P. falciparum* and decreased activity against human ubiquitin-activating enzyme (UAE). ML471 exhibits enhanced cellular and biochemical selectivity. It also exhibits activity against plasmodium liver stages and sexual transmissible stages. Importantly, ML471 exhibits rapid killing kinetics and demonstrates single-dose oral efficacy against *P. falciparum* in an *in vivo* model.

## Results and discussion

### Potency and selectivity of ML901 derivatives

We examined the activity of a series of pyrazolopyrimidine sulfamates with different substitutions at the 7-position (Fig 1A–1I), from the Takeda Pharmaceuticals Library, against the growth of asexual blood stage *P. falciparum* (3D7 strain) in a 72-h exposure assay. Consistent with our previous report [16], ML901 exhibits potent activity with a 50% Inhibitory Concentration (IC$_{50\_72h}$) of 2.8 nM (Table 1). Indeed, most of the compounds show excellent potency (Table 1), demonstrating that different substitutions at the 7-position are well-tolerated, except for the bulky phenoxy substituent (ML470) (IC$_{50\_72h}$ = 44.8 nM; Table 1). The non-specific inhibitor, adenosine 5'-monosulfamate (AMS, Fig 1J [16]), which has a different heterocyclic base and lacks the 7-position substituent, also exhibits potent activity (IC$_{50\_72h}$ = 1.8 nM; Table 1). ML471, which bears an isopropyl group at the 7-position, exhibits very potent activity (IC$_{50\_72h}$ = 1.5 nM, Table 1).

Short duration pulse exposure to antimalarial drugs can be used to dissect differences in potency of compounds against different stages of development that may not be evident in standard 72-h exposure assays [18]. Here we subjected tightly synchronised blood stage *P. falciparum* cultures to different duration pulses of ML901 and measured parasitemia in the next cycle [19]. Schizont stage parasites are efficiently killed by ML901, even when exposed to pulses as short as 3 h or 6 h (S1B Fig and S1 Table). This may reflect the need for synthesis of daughter merozoite proteins during schizogony. By contrast, trophozoite stage parasites are 10 to 20-fold less sensitive (S1A Fig and S1 Table).

To determine whether ML471 and other selected pyrazolopyrimidine sulfamates exhibit enhanced potency, we exposed synchronised trophozoite stage cultures to 6-h pulses and measured growth inhibition by quantifying the SYBR Green I fluorescence signal in the next cycle.

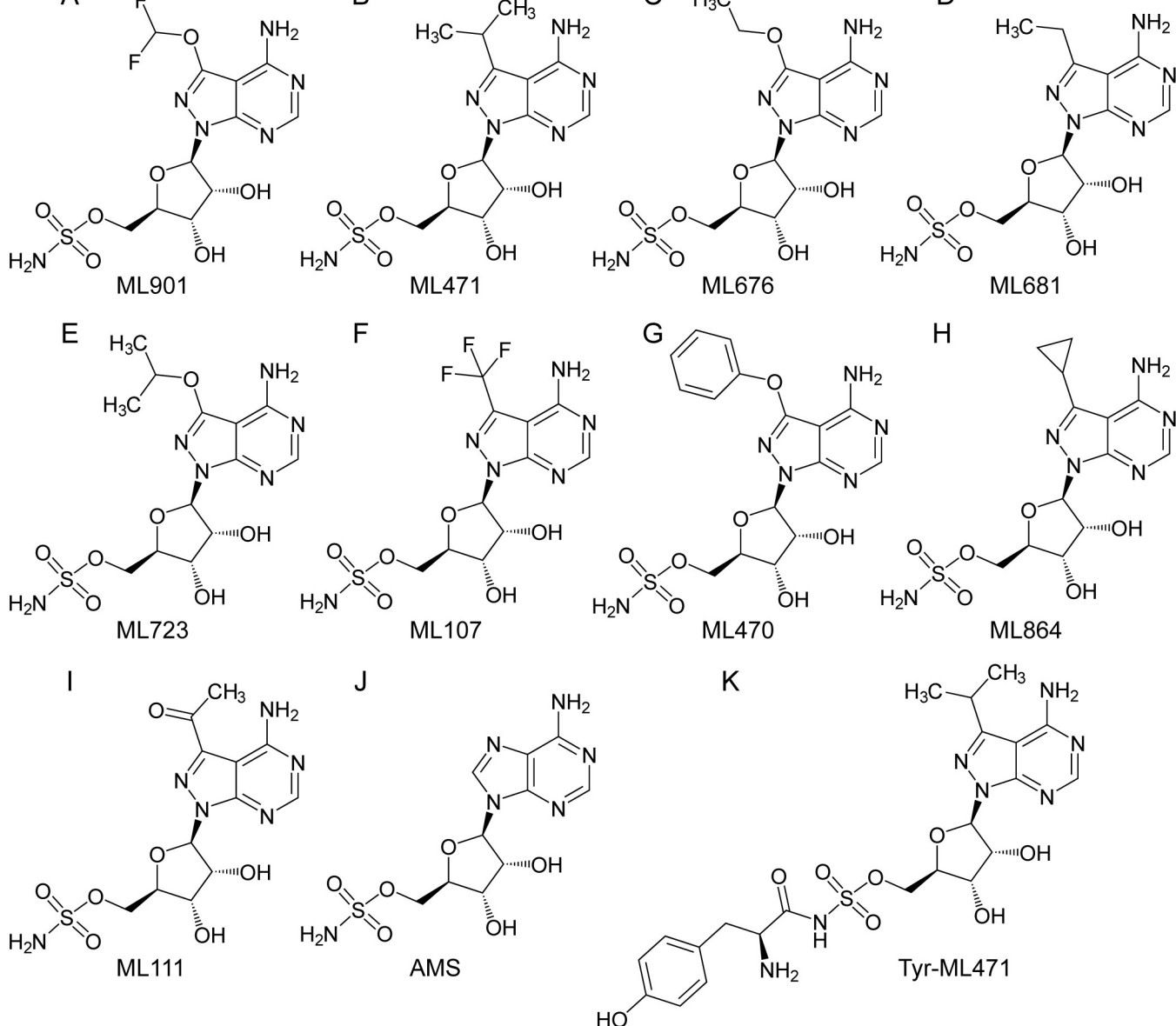

**Fig 1. Structures of ML901 and derivatives and adenosine 5'-sulfamate (AMS).** (A) ML901, (B) ML471, (C) ML676, (D) ML681, (E) ML723, (F) ML107, (G) ML470, (H) ML864, (I) ML111, (J) AMS, (K) Tyr-ML471.

ML471 exhibits enhanced potency ($IC_{50\_6h}$ = 29.1 nM, Fig 2A and Table 1) compared with ML901 and the other analogues tested (Fig 2A and Table 1, $IC_{50\_6h}$ values ranging from 135 nM to 220 nM).

We examined the toxicity of the compounds against the human HepG2 cell line, employing a 72-h exposure period. ML901 inhibits the growth of HepG2 cells with an $IC_{50\_72h}$ of 4.65 μM (Table 1). Some of the compounds from the pyrazolopyrimidine sulfamate series, including ML471 exhibited markedly improved selectivity, with $IC_{50}$ values above the range of the assay (>50 μM, Table 1). Other compounds such as ML107, which has a trifluoromethyl substituent, and ML681 and ML864 which have slightly smaller substituents as compared to an isopropyl group, show a higher level of toxicity against the mammalian cell line (Table 1). These data

**Table 1. Activities of pyrazolopyrimidine sulfamates as inhibitors of parasite growth.** *P. falciparum* (3D7, ring stage) cultures were exposed to different nucleoside sulfamates for 72 h and the 50% Inhibitory Concentration ($IC_{50}$) for growth inhibition assessed using the lactate dehydrogenase (*Pf*LDH) assay. Alternatively, synchronized Cam3.II[rev] parasite cultures were subjected to 6-h pulses of nucleoside sulfamates, at the trophozoite (25–30 h.p.i.) stage. 50% growth inhibition ($IC_{50}$) was determined in the cycle following treatment, using a SYBR Green I assay. HepG2 cell cultures were exposed to nucleoside sulfamates for 72 h and growth inhibition ($IC_{50}$) assessed using CellTiter-Glo reagent. Data represent the mean of three independent experiments and error bars correspond to SEM. AMS = Adenosine 5'-sulfamate. n = Number of biological repeats. Medicines for Malaria Venture (MMV) designations for the compound names are in brackets.

| Compound | *P. falciparum* | | *H. sapiens* (HepG2) |
|---|---|---|---|
| | 72-h $IC_{50}$ (nM) (3D7) | 6-h Trophozoite Stage $IC_{50}$ (nM) (Cam3.II) | 72-h $IC_{50}$ (nM) |
| **ML901** (MMV1581329) | 2.8 ± 0.2 (n = 3) | 135 ± 14 (n = 3) | 4,650 ± 1,390 (n = 3) |
| **ML471** (MMV1793207) | 1.5 ± 0.2 (n = 3) | 29.1 ± 3.0 (n = 3) | > 50,000 (n = 3) |
| **ML676** (MMV1793313) | 2.8 ± 0.2 (n = 3) | N/A | 22,400 ± 7,300(n = 3) |
| **ML681** (MMV1793314) | 3.4 ± 0.1(n = 3) | N/A | 7,530 ± 1,250 (n = 3) |
| **ML723** (MMV1793208) | 2.5 ± 0.6 (n = 3) | 220 ± 31 (n = 3) | > 50,000 (n = 3) |
| **ML107** (MMV1793318) | 4.2 ± 0.5 (n = 3) | 150 ± 33 (n = 3) | 2,520 ± 820 (n = 3) |
| **ML470** (MMV1793342) | 44.8 ± 8.2 (n = 4) | N/A | > 50,000 (n = 3) |
| **ML864** (MMV1793301) | 2.5 ± 0.6 (n = 3) | N/A | 21,300 ± 1,700 (n = 3) |
| **ML111** (MMV1793328) | 4.3 ± 0.7 (n = 3) | N/A | > 50,000 (n = 3) |
| **AMS** | 1.8 ± 0.6 (n = 3)* | N/A | N/A |

* Data from [16]

suggest that the size of the 7-position substituent impacts selectivity. Our previous report [16] showed that AMS is also toxic to mammalian cells lines, such as HCT116 ($IC_{50\_72h}$ = 26 nM), in agreement with previous reports [20, 21], and, as expected, given its broad inhibitory activity.

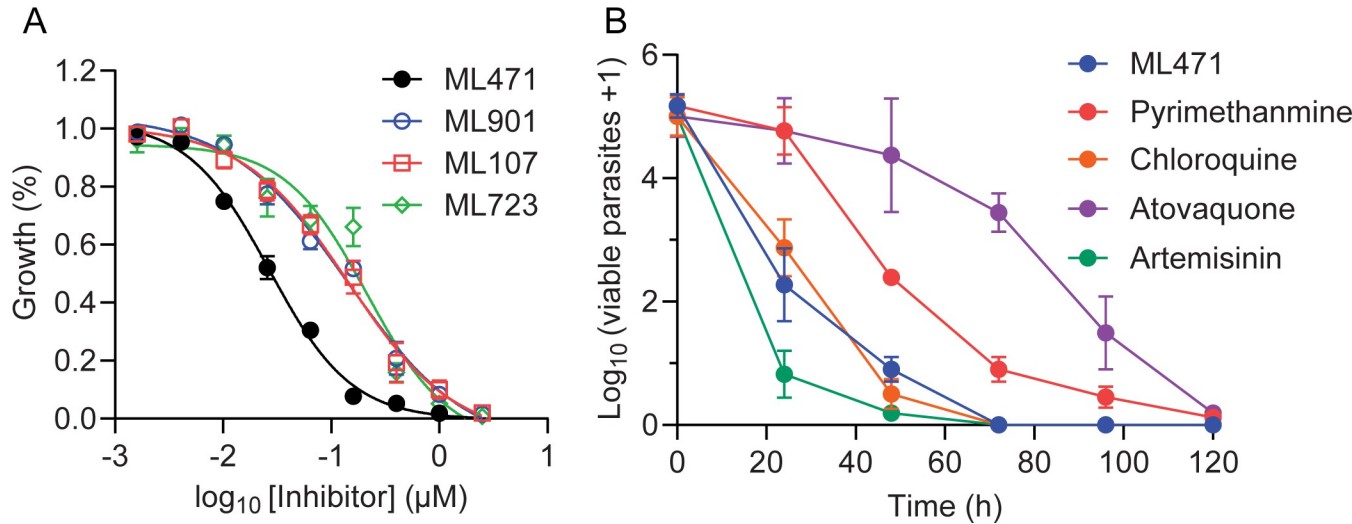

**Fig 2. ML471 exhibits improved short-exposure activity against *P. falciparum* cultures, associated with rapid parasite killing.** (A) Synchronized Cam3.II[rev] parasite cultures were subjected to 6-h pulses of ML901, ML471, ML107 and ML723, at the trophozoite (25–30 h.p.i.) stage. Growth inhibition was determined in the cycle following treatment. Data represent the mean of three independent experiments and error bars correspond to SEM. (B) 3D7 parasite cultures were treated for 0 to 120 h with ML471 or compounds with fast (artemisinin, chloroquine), moderate (pyrimethamine) or slow (atovaquone) killing profiles, at 10 times their respective $IC_{50\_48h}$ values. Following removal of inhibitor, serial dilutions of cultures were established, and assessed after 18 days of culturing.

## ML471 exhibits low activity against critical E1 enzymes

ML901 was originally investigated as an inhibitor of Atg7, an E1 that activates the Autophagy-related protein 8 (Atg8), a ubiquitin-like protein involved in the formation of autophagosomal membranes [12,22]. However, Atg7 is not essential for cell survival *in vitro* [23] and is unlikely to underpin the observed mammalian cell toxicity [12]. By contrast, loss-of-function of other E1 enzymes, in particular UAE, is known to be deleterious to the survival and growth of cells [10,11,24,25].

To explore the molecular basis of the enhanced cellular selectivity of ML471 and other derivatives compared with ML901, we assessed inhibitory activity against a range of E1 enzymes. ML901 exhibits strong inhibition of Atg7 ($IC_{50}$ = 33 nM) and clinically relevant activity against UAE ($IC_{50}$ = 5.39 μM), with lower-level activity against NEDD8 Activating Enzyme (NAE $IC_{50}$ = 28 μM) and no activity against SUMO Activating Enzyme (SAE) (Table 2). ML676, ML723, ML681, ML107 and ML111 also exhibit relevant activity against one or more of UAE, NAE or SAE, consistent with their weak to moderate cellular toxicity (Table 2). Such off-target activity could limit the development of these compounds. As previously reported [16], AMS is a potent inhibitor of each of the E1s tested (Table 2), likely contributing to the high cellular toxicity. ML471 inhibits the activity of human Atg7 ($IC_{50}$ = 22 ± 9 nM, Table 2), but exhibits no or very little activity against UAE, NAE and SAE, consistent with low mammalian cell cytotoxicity.

## Parasitological properties of ML471

Given its enhanced potency and selectivity, ML471 was selected for further characterisation. In addition to potent and selective activity against laboratory asexual blood stage *P. falciparum*, new antimalarial compounds should exhibit activity against clinical strains of *P. falciparum* and *P. vivax* and, preferably, exhibit activity against liver and transmissible stages. ML471 exhibits potent activity against South American clinical isolates of *P. falciparum* and *P. vivax*, including chloroquine-resistant strains, with median $IC_{50}$ values of 4.2 nM (*Pf*) and 8.0 nM (*Pv*), respectively, similar to artesunate (S2 Fig and S2 Table). ML471 exhibits improved potency compared with ML901 against gametocytes at both the early ($IC_{50}$ = 112 nM) and mature ($IC_{50}$ = 392 nM) stages of development, with potencies similar to those for Methylene Blue and the *Plasmodium* phosphatidylinositol 4-kinase (PI4K) inhibitor, MMV390048 (S3

**Table 2. Inhibitory activity of selected pyrazolopyrimidine sulfamates in E1 enzyme assays.** Homogeneous Time-Resolved Fluorescence (HTRF) enzyme assays were employed to evaluate the 50% Inhibitory Concentrations ($IC_{50}$) values for compounds against Atg7 (autophagy-related protein-7), NAE (NEDD8-activating enzyme), UAE (ubiquitin activating enzyme) and SAE (SUMO-activating enzyme) with appropriately tagged ubiquitin-like proteins and E2 conjugating enzymes as described in the Methods. GABARAP = GABAA receptor-associated protein. Data represent mean ± SEM. n = Number of independent experiments.

| Compound | ATG7 IC$_{50}$ HTRF RH-GABARAP (μM) | NAE IC$_{50}$ HTRF (μM) | UAE IC$_{50}$ HTRF (μM) | SAE IC$_{50}$ HTRF (μM) |
|---|---|---|---|---|
| **ML901** | 0.033 ± 0.003 (n = 57)* | 28.0 ± 0.6 (n = 3) | 5.39 ± 0.160 (n = 3) | >100 (n = 3) |
| **ML471** | 0.022 ± 0.009 (n = 6) | >100 (n = 3) | 85.7 ± 6.5 (n = 3) | >100 (n = 3) |
| **ML681** | N.A. | 35.3 ± 0.8 (n = 3) | 8.7 ± 0.6 (n = 3) | 22.4 ± 3.1 (n = 3) |
| **ML676** | N.A. | 66.6 ± 1.1 (n = 3) | 73.4 ± 3.5 (n = 3) | 31.4 ± 5.7 (n = 3) |
| **ML723** | N.A. | >100 (n = 3) | >100 (n = 3) | >100 (n = 3) |
| **ML470** | N.A. | >100 (n = 3) | >100 (n = 3) | >100 (n = 3) |
| **ML107** | N.A. | 75.3 ± 0.8 (n = 3) | 22.9 ± 0.7 (n = 3) | 78.5 ± 6.2 (n = 3) |
| **AMS*** | 410 ± 20 (n = 90) | 0.006 ± 0.001 (n = 9) | 0.006 ± 0.003 (n = 3) | 0.006 ± 0.002 (n = 7) |

*Data from [16]

Fig and S3 Table). The lower potency against gametocytes compared with the asexual blood stage may reflect the fact that gametocytes develop more slowly and are therefore less susceptible to inhibition of protein translation. ML471 prevents development of both *P. falciparum* NF175 and NF135 schizonts in primary human hepatocytes with high potency (IC$_{50}$ = 2.8 nM for NF175 and IC$_{50}$ = 5.5 nM for NF135, S4 Fig and S4 Table), while exhibiting no toxicity against the primary hepatocyte host cells (S4 Table). ML471 potently inhibits the fertility of transmissible male (IC$_{50}$ = 49 nM) and female (IC$_{50}$ = 260 nM) gametocytes (S5 Fig and S5 Table). The positive control, Cabamiquine, exhibited potent activity against both gametocyte sexes (S5 Fig and S5 Table), consistent with a previous report [26]. In each of these assays, ML471 exhibits similar or improved potency compared with ML901 (S3–S5 Figs and S3–S5 Tables).

The Parasite Reduction Rate (PRR) was assessed using a standardized method [27] and compared with compounds exhibiting very fast (artemisinin), fast (chloroquine), moderate (pyrimethamine) or slow (atovaquone) killing profiles, at 10 times their respective IC$_{50\_48h}$ values. The Log PRR for ML471 of 4.1 is considered fast, and is similar to chloroquine (Fig 2B and S6 Table).

## Pharmacological properties of ML471

To meet MMV candidate selection criteria new antimalarial compounds for treatment indications would minimally need to have an oral dose of <500 mg to achieve a 12-log kill in a 55 kg adult. ML471 exhibits a favourably low molecular weight (MW = 388) and good solubility (S7 Table). It has a predicted AlogP of -1.18 and a Topological Polar Surface Area (TSPA) of 189 Å$^2$ (S7 Table), which suggest this compound may have difficulty being absorbed but should be metabolically stable. As expected, rat oral bioavailability needs optimisation (%F (blood) = 8.72 to 9.56, n = 3, S7 Table) and renal clearance of the parent compound was evident (9–38% of the dose recovered in 0–24 h urine as parent, S7 Table). Despite the sub-optimal oral absorption, the rat pharmacokinetic profile of ML471 (25 mg/kg p.o.; Fig 3A and S7 Table) exhibits excellent duration of absorbed drug exposure. The area under the curve is 30 μM.h, reflecting the low blood clearance (~4% of liver blood flow after an IV dose of 1 mg/kg) and the long terminal half-life in blood (T$_{1/2\infty}$ = 30.5 h) (S7 Table). Acting in its favour, ML471 shows high retention in red blood cells (RBCs) in the i.v. PK study, with blood to plasma ratios around 1 at the initial sampling times, but increasing greatly over time due to slower clearance from the RBC compartment (S7 Table; and compare Fig 3A and 3B). ML471 contains a sulfamate group that is predicted to bind tightly to carbonic anhydrase [28, 29], which likely explains accumulation of ML471 into RBCs (where carbonic anhydrase is abundant and where the asexual stage parasites are located).

## Efficacy of ML471 in a SCID model of *P. falciparum*

Single, low-dose oral efficacy is a key requirement for new antimalarial treatments to be used as part of the MMV's Single Encounter Radical Cure and Prophylaxis (SERCAP) target product profile (TPP1) [8]. We determined the *in vivo* antimalarial efficacy of ML471 in severe combined immune deficient (SCID) mice, engrafted with human RBCs and infected with *P. falciparum* [30,31], which is the gold standard for testing *in vivo* efficacy of malaria drug candidates. Oral dosing results in good exposure. Single doses of 100 mg/kg p.o. and 200 mg/kg p.o. resulted in AUC$_{0-120h}$ values of 640 μM.h (250,000 h.ng/mL) and 550 μM.h (215,000 h.ng/mL), respectively (Fig 3C and S8 Table), while a dosing regimen of 4 x 50 mg/kg p.o. resulted in an area under the curve (AUC$_{0-24h}$) of 54 μM.h (21,100 h.ng/mL), assessed at 24 h, after the first

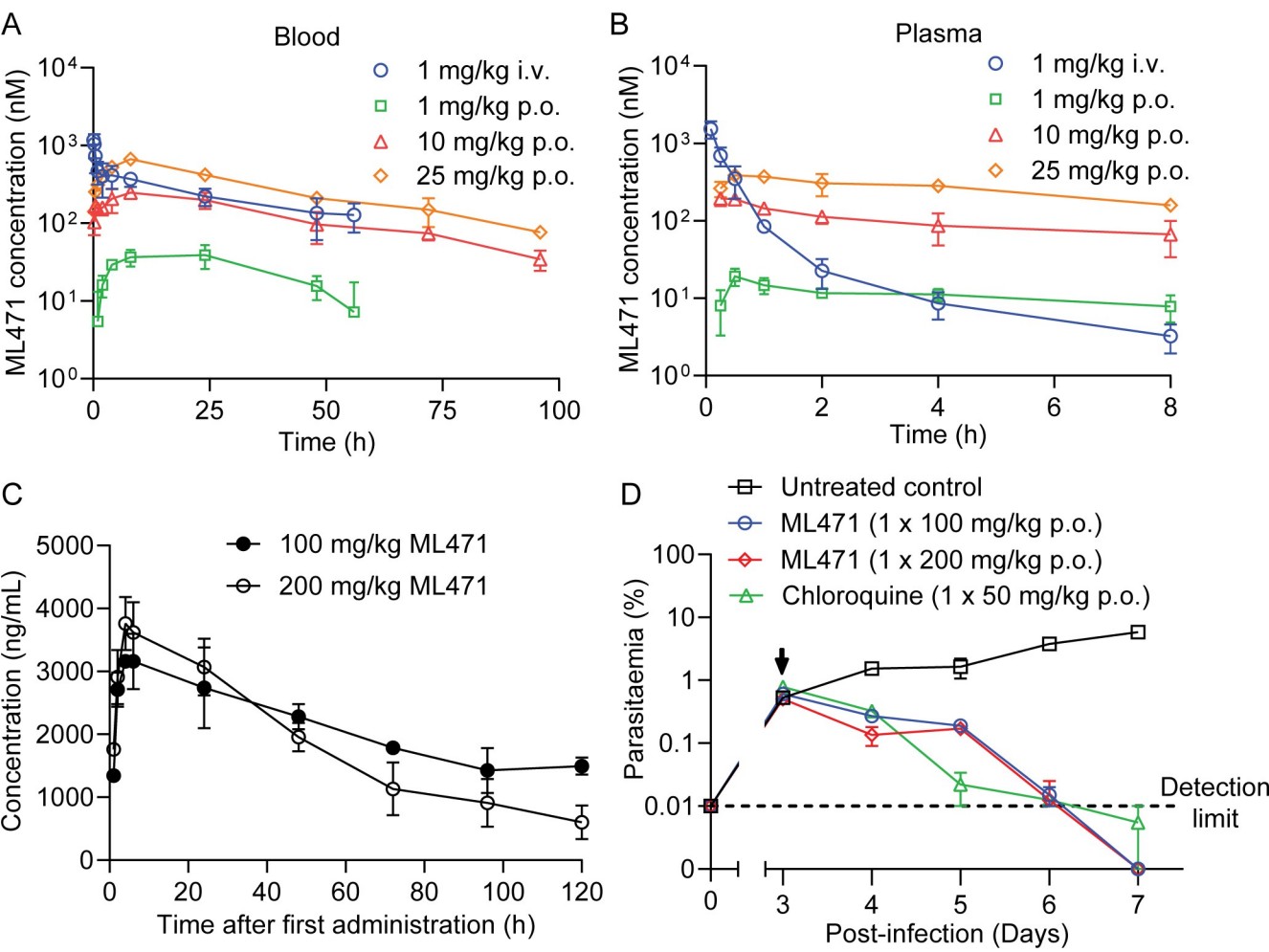

**Fig 3. Pharmacokinetics profiles and *in vivo* efficacy of ML471.** (A, B) Rat pharmacokinetics for ML471. Rats were dosed with ML471 at 1 mg/kg i.v. (blue) or 1, 10 or 25 mg/kg p.o. (green, red, orange), and blood (A) and plasma (B) samples were collected for analysis. See S7 Table for pharmacokinetics values. (C) Pharmacokinetics profile (in blood), for SCID mice engrafted with human RBCs infected with *P. falciparum*, over the first day following treatment with ML471 at 100 or 200 mg/kg p.o.. See S8 Table for pharmacokinetics values. (D) Therapeutic efficacy of ML471 in the SCID mouse *P. falciparum* model, dosed with ML471 at 100 or 200 mg/kg p.o. on Day 3 post-infection (arrowed). The chloroquine data are from [16].

dose (S6 Fig and S8 Table). All doses were well tolerated, and the long half-life and high exposure are encouraging as these are important properties for single dose antimalarials.

Mice were infected intravenously with $2 \times 10^7$ *P. falciparum* (*Pf*3D7$^{0087/N9}$) and ML471 was administered on day 3 post-infection. The single dose regimen (either 100 or 200 mg/kg p.o.) was sufficient to achieve reduction of 3D7 parasitemia to baseline, with a parasite clearance rate similar to that of chloroquine (CQ; 50 mg/kg p.o.) (Fig 3D) and no evidence of toxicity. Similarly, the dosing regimen of 4 x daily doses of 50 mg/kg p.o. reduced the 3D7 parasitemia to baseline with a clearance rate similar to chloroquine (CQ; 4 x 50 mg/kg p.o.) (S6B Fig).

## ML471 and ML901 selection leads to amplification of the *Pf*TyrRS locus

*In vitro* evolution of resistance, under a standardized protocol, has been used to assess the propensity for the development of resistance [32–34]. Here, we examined the resistance potential of both ML901 and ML471, employing a single-step selection with Dd2-B2 parasites. For ML901, the parasites were subjected to pressure at 3 x IC$_{50}$, while for ML471, parasites were

subjected to 10 x $IC_{50}$. With both compounds, parasites were retrieved and $IC_{50}$ shifts were observed (ranging from two- to 16-fold; S9 Table). The Minimum Inoculum for Resistance (MIR) values for ML901 and ML471 were estimated to be $10^7$ and 7.1 x$10^5$, respectively (S9 Table). These values are at or below the preferred threshold for further development, making this a parameter of concern. Use of these sulfamates in a drug combination could suppress the evolution of resistant mutants.

We performed whole-genome sequencing of parasite lines selected for resistance. Copy number variations (CNVs) were found in flasks selected with either ML901 or ML471, with amplifications always containing the *Pf*TyrRS gene located within amplicons of varying sizes (S10 and S11 Tables). This gene is present in 2–4 copies in the amplified lines. This finding is consistent with our earlier identification of *Pf*TyrRS as the target of ML901 [16]. No SNPs were found in any of the samples, in contrast to a previous report [16]. This may be due to the slow ramp-up exposure method employed in the earlier study.

## ML471 targets *P. falciparum* tyrosine tRNA synthetase via a reaction hijacking mechanism

Reaction hijacking inhibition of *Pf*TyrRS is expected to lead to the formation of Tyr-ML471 adducts (Fig 1K) in the active site. We treated *P. falciparum* infected RBCs for 2 h with 1 μM ML471 and subjected extracts to LC-MS to search for amino acid-ML471 conjugates. An LC-MS peak corresponding to Tyr-ML471 precursor ion (*m/z* 552.1871) was detected at the retention time of 3.0 min (Fig 4A). Synthetic Tyr-ML471 was generated as a standard to confirm the peak assignment (Figs 4A and S7A). None of the other 19 possible amino acid conjugates were detected.

Recombinant *Pf*TyrRS was produced in *Escherichia coli* as previously described [16]. To examine the ability of *Pf*TyrRS to generate the Tyr-ML471 conjugate, the enzyme was incubated with ATP, Tyr, tRNA$^{Tyr}$ and ML471. Following sample extraction, LC-MS analysis revealed a peak at *m/z* 552.1868 with a retention time of 3.0 min, consistent with that of the Tyr-ML471 precursor ion (S7B Fig). MS/MS analysis further confirmed the identity of the conjugate (S7C Fig).

## Recombinant *P. falciparum* tyrosine tRNA synthetase is thermally stabilised upon formation of the Tyr-ML471 adducts

When ML471 was incubated with *Pf*TyrRS in the presence of all other substrates (*i.e.*, Tyr, ATP and *Pf*tRNA$^{Tyr}$), the apparent protein melting point ($T_m$), measured by differential scanning fluorimetry (DSF), increased by a remarkable 18°C (Fig 4B and Tables 3 and S12). The increase in thermal stability is even greater than that induced by the Tyr-ML901 adduct (Tables 3 and S12), which is consistent with the enhanced antimalarial activity of ML471 compared with ML901. Importantly, when incubated in the presence of ML901 or ML471, and all substrates, recombinant *Hs*TyrRS was not stabilised (Fig 4C, red and orange curves). This shows that the human enzyme is not susceptible to hijacking by ML471. By contrast, incubation of *Hs*TyrRS with the broad specificity compound, AMS, and all substrates, leads to substantial thermal stabilization, consistent with efficient reaction hijacking by AMS (Fig 4C and Tables 3 and S12).

## ML471 inhibits ATP consumption by *Pf*TyrRS

Recombinant *Pf*TyrRS consumes ATP at a moderate level, even in the absence of tRNA, due to the generation and release of AMP-Tyr in the initial reaction phase (Fig 4D). The rate of ATP consumption is increased 6-fold upon addition of tRNA$^{Tyr}$, consistent with productive

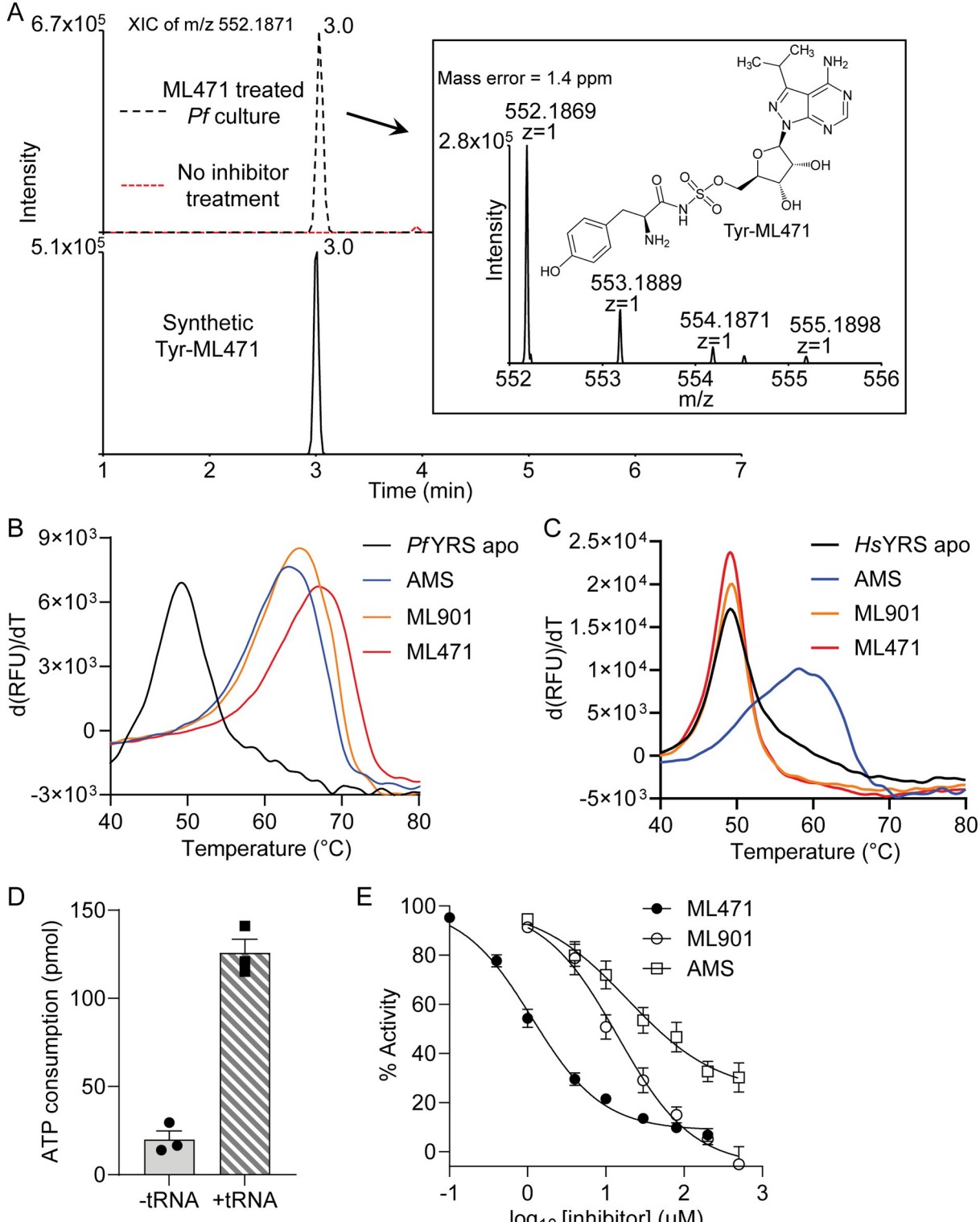

**Fig 4. Identification of ML471 conjugates in *P. falciparum* and effects of nucleoside sulfamates on enzyme stability and activity.** *P. falciparum*-infected RBCs were treated with 1 μM ML471 for 2 h. Extracts were subjected to LCMS and the expected mass for amino acid-ML471 conjugates searched. (A) Upper panel shows the extracted ion chromatograms of the anticipated Tyr-ML471 adduct at *m/z* 552.1871 extracted from ML471 treated *P. falciparum* culture (black trace) and untreated control (red trace). Lower panel shows the synthetic Tyr-ML471 conjugate at 0.2 μM. The inset shows the MS analysis of the parasite-generated Tyr-ML471, and the structure of Tyr-ML471. Profiles

are typical of data from 3 independent experiments. (B,C) First derivatives of melting curves for *Pf*TyrRS (B) and *Hs*TyrRS (C) (2.3 μM) in the apo form or after incubation at 37˚C with ML901, ML471 or AMS, in the presence of 10 μM ATP and 20 μM tyrosine. For *Pf*TyrRS, 50 μM nucleoside sulfamate and 4 μM *Pf*tRNA$^{Tyr}$ were incubated with substrates for 2 h. For *Hs*TyrRS, 200 μM nucleoside sulfamate and 8 mg/mL yeast tRNA were incubated with substrates for 4 h. Data are representative of three independent experiments. (D) ATP consumption by *Pf*TyrRS in the presence and absence of the cognate tRNA$^{Tyr}$. ATP consumption in the absence of tRNA$^{Tyr}$ derives from turnover of Tyr-AMP generated in the initial phase of the TyrRS reaction. The reaction component concentrations are: *Pf*TyrRS (25 nM), ATP (10 μM), tyrosine (200 μM), pyrophosphatase (1 unit/mL) and cognate tRNA$^{Tyr}$ (4.8 μM), if present; and incubations were at 37˚C for 1 h. Data are the average of three independent experiments and error bars correspond to SEM. (E) Effects of increasing concentrations of ML471, ML901 and AMS on ATP consumption by *Pf*TyrRS. Assay conditions are the same as in (D), with cognate tRNA$^{Tyr}$. Data represent mean ± SEM from three or four independent experiments.

aminoacylation. ML471 inhibited ATP consumption by *Pf*TyrRS when added in the presence of *Pf*tRNA$^{Tyr}$ (IC$_{50}$ = 1.4 μM) much more potently than ML901 (IC$_{50}$ = 13.4 μM) (Fig 4E and Table 3). Indeed, ML471 is 6 to 30 times more effective than the other ML901 analogues examined (S8A and S8B Fig and Table 3). AMS is also significantly less potent in this assay (IC$_{50}$ = 51.7 μM) (Fig 4E and Table 3). The concentration of ML471 needed to induce 50% inhibition of ATP consumption (1.4 μM) is much higher than the amount needed to kill parasites in a 72-h exposure assay (1.5 nM). In the biochemical assay, the recombinant enzyme generates a charged tRNA product, which is then attacked by ML471 to generate the inhibitory Tyr-ML471 adduct. By contrast, in cells, tRNAs are generally fully loaded [35], which is expected to promote adduct formation. In addition, binding of sulfamate-containing compounds to carbonic anhydrase [28,29] will facilitate accumulation of ML471 into infected red blood cells.

## Docking of ML471 into UAE reveals differential interactions within the active site

Susceptibility to reaction hijacking depends on the ability of the nucleoside sulfamate to bind in the ATP-binding pocket of the relevant adenylate-forming enzyme, in a pose that is suitable

**Table 3. Activities of pyrazolopyrimidine sulfamates in selected biochemical assays.** ATP consumption by *Pf*TyrRS (25 nM) in the presence of ATP (10 μM), tyrosine (200 μM), pyrophosphatase (1 unit/mL) and cognate tRNA$^{Tyr}$ (4.8 μM), was measured using the Kinase Glo assay after incubation at 37˚C for 1 h. The T$_m$ values for *Pf*TyrRS and *Hs*TyrRS (2.3 μM) were measured in the apo form or after incubation at 37˚C for 2 h (*Pf*TyrRS) or 4 h (*Hs*TyrRS) with the nucleoside sulfamates (50 μM with *Pf*TyrRS and 200 μM with *Hs*TyrRS) in the presence of 10 μM ATP, 20 μM tyrosine, 4 μM cognate tRNA$^{Tyr}$ (*Pf*TyrRS) or 8 mg/mL yeast tRNA (*Hs*TyrRS). *K$_D$ values (apparent) are estimated from differential scanning fluorimetry (DSF) analysis using an irreversible protein thermal unfolding model that has been described previously [59]. Data values represent mean ± SEM. AMS = Adenosine 5'-sulfamate. n = Number of biological repeats.

| Compound | Inhibition of *Pf*TyrRS (Kinase Glo) | *Pf*TyrRS binding (DSF) (n = 3) | | *Hs*TyrRS binding (DSF) (n = 3) | |
|---|---|---|---|---|---|
| | IC$_{50}$ (μM) | Delta T$_m$ (˚C) | apparent K$_D$* (x10$^{-9}$) (M) | Delta T$_m$ (˚C) | apparent K$_D$* (x10$^{-9}$) (M) |
| ML901 | 13 ± 3 (n = 3) | 15.2 ± 0.1 | 0.7 | 0.4 ± 0.1 | N/A |
| ML471 | 1.4 ± 0.2 (n = 3) | 18.0 ± 0.1 | 0.2 | 0.06 ± 0.04 | N/A |
| ML676 | 48 ± 16 (n = 5) | 15.69 ± 0.01 | 0.6 | N/A | N/A |
| ML681 | 23 ± 8 (n = 3) | 16.4 ± 0.1 | 0.4 | N/A | N/A |
| ML723 | 14 ± 5 (n = 3) | 16.2 ± 0.1 | 0.5 | N/A | N/A |
| ML107 | 24 ± 5 (n = 3) | 13.6 ± 0.6 | 1.4 | N/A | N/A |
| ML470 | 41 ± 23 (n = 4) | 15.7 ± 0.3 | 0.6 | N/A | N/A |
| ML864 | 8.5 ± 4.5 (n = 4) | 17 ± 0.1 | 0.3 | N/A | N/A |
| ML111 | 34 ± 14 (n = 3) | 15.1 ± 0.2 | 0.7 | N/A | N/A |
| AMS | 52 ± 16 (n = 4) | 13.6 ± 0.7 | 1.3 | 9.1 ± 0.1 | 8.0 |

for reaction with the relevant enzyme-bound product. To probe the molecular basis for the enhanced selectivity of ML471 compared with ML901, we used the Surflex-Dock molecular docking module in SybylX2 to dock ML901 and ML471 into the ATP-binding site of human UAE (6DC6) [36]. For comparison, ML901 and ML471 were docked into the binding pockets of the A- and B-chains of the *Pf*TyrRS/ Tyr-ML901 complex (7ROS) [16], noting that the two chains of the dimeric *Pf*TyrRS structure in 7ROS show differences in the positions of key residues around the Tyr-ML901 ligand that are thought to relate to altered mobility of the KMSKS loop [16].

As described above, ML901 and ML471 bear, respectively, difluoromethoxy and isopropyl groups at the 7-postion of the pyrazolopyrimidine ring. Both ML901 and ML471 can be docked into the active sites of the A- and B-chains of *Pf*TyrRS, with the 7-position substituent located in a solvent accessible pocket. Fig 5A illustrates the B-chain with docked ML901 (aqua carbons), overlaid with the pose adopted when ML901 (yellow carbons) is docked into the A-chain. In both cases, the difluoromethoxy group is positioned away from His70 (red arrow). By contrast, when ML471 (aqua carbons) is docked into the B-chain and overlaid with the pose adopted when ML471 (yellow carbons) is docked into the A-chain, the isopropyl group is positioned closer to His70 (Fig 5B), indicative of a clash.

Overlay of the docked conformations of ML901 and ML471 with the published *Pf*TyrRS/ Tyr-ML901 structure (7ROS) shows close alignment of the nucleoside sulfamate regions (7ROS) (Fig 5C and 5D). Again, the different poses of the difluoromethoxy and isopropyl groups are evident (red arrows). Of interest, the ribose ring of Tyr-ML901 in the crystal structure is twisted relative to the docked nucleoside sulfamates (purple arrows). This may arise from the conjugation of the ML901 sulfamate to tyrosine, which repositions the ligand (Fig 5C and 5D).

When ML901 and ML471 are docked into the ATP-binding site of human UAE, the oxygen of the ML901 difluoromethoxy group makes a H-bond interaction with Asn577 (Fig 5E) that cannot exist for the isopropyl group in ML471 (Fig 5F). Instead, the hydrophobic isopropyl group must pack into a polar space flanked by Asn577 and Arg551, an unfavourable interaction that is likely to reduce the affinity of ML471 for the site. The sulfamate groups are positioned slightly differently (Fig 5E and 5F). However, as the chemical structure of this region of the two compounds is identical, this difference is regarded as an artefact of the docking rather than a significant alteration of the probable binding pose. Thus, differences in the interactions made by the 7-position substituent appear likely to underpin the improved selectivity of ML471.

## Crystal structure of *Pf*TyrRS-Tyr-ML471 provides insights into the molecular basis for enhanced potency

The complex of *Pf*TyrRS with synthetic Tyr-ML471 was crystalised using previously established conditions [16] and the structure refined at 1.8 Å resolution (S13 Table), revealing a homodimer organization (Fig 6A) with clear density for the Tyr-ML471 ligand. As described previously [16], *Pf*TyrRS is a Class I aaRS, characterized by a catalytic domain with a Rossmann fold (residues 18–260) linked to a C-terminal anti-codon binding domain (residues 261–370) involved in recognition of tRNA$^{Tyr}$. *Pf*TyrRS contains "HIGH" and "KMSKS" ($_{70}$HIAQ$_{74}$ and $_{247}$KMSKS$_{251}$ in *Pf*TyrRS) motifs that are characteristic of the catalytic domain of Class I tRNA synthetases (sub-class c).

Tyr-ML471 makes many interactions with active site residues, involving the pyrazolopyrimidine amine, ribose hydroxyls, sulfamate, and tyrosine (Figs 6B, S9A and S9B). The 7-position isopropyl group is oriented away from the binding pocket and is partially solvent exposed,

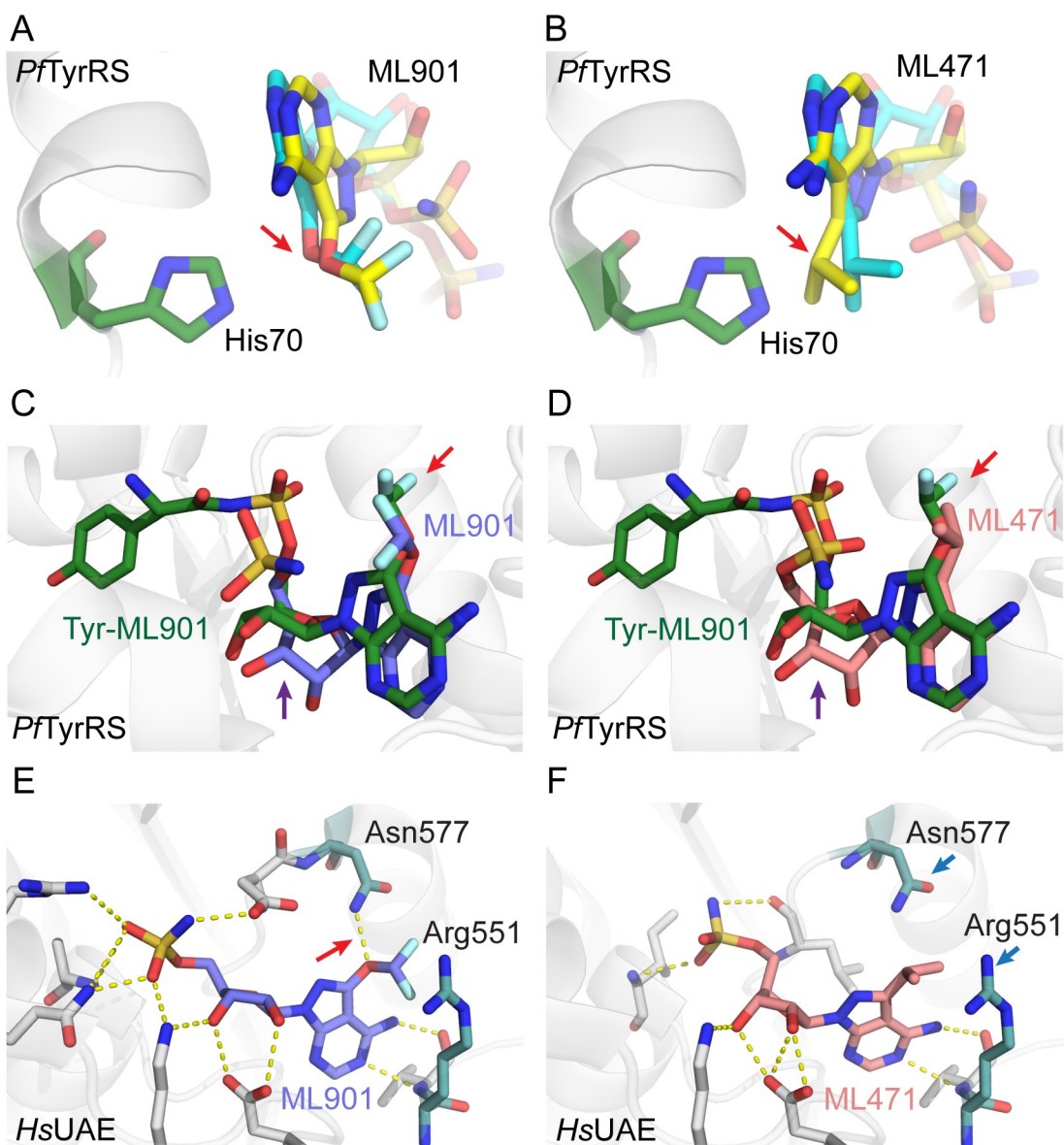

**Fig 5. Docking of ML901 and ML471 into structures of *Pf*TyrRS and UAE provides insights into selectivity.** (A) Active site of *Pf*TyrRS/Tyr-ML901 (7ROS) B-chain (His70 depicted in green) with docked ML901 (aqua carbons). The model is overlayed with ML901 (depicted with yellow carbons) with the pose adopted upon docking into the A-chain. (B) Active site of *Pf*TyrRS/Tyr-ML901 (7ROS) B-chain (His70 depicted in green) with docked ML471 (aqua carbons). The model is overlayed with ML471 (depicted with yellow backbone) with the pose adopted upon docking into the A-chain. The red arrow illustrates the different conformations adopted by the difluoromethoxy and isopropyl groups. (C, D) The structure of 7ROS B-chain with bound Tyr-ML901 is overlayed with B-chain-docked ML901 (C) and ML471 (D). The red arrows illustrate the different conformations adopted by the difluoromethoxy and isopropyl groups. The purple arrows illustrate the twisted ribose group in the Tyr-ML901 conjugate. By contrast, in the docked nucleoside sulfamates, the ring systems are co-planar. (E,F) ML901 (E) and ML471 (F) were docked into the ATP-binding site of human UAE (6DC6). A H-bond made by ML901 with residue Arg 551 is indicated with a red arrow. Asn577 and Arg551 (blue arrows) flank the hydrophobic isopropyl group in the ML471 dock.

consistent with the docking study. The isopropyl group of ML471 is oriented differently in the A- and B-chains and the adjacent His70 (of $_{70}$HIAQ$_{73}$) adopts different side chain rotamers in each chain (Figs 6C and S9C). The majority of the KMSKS loop was poorly defined in the electron density in both chains, suggesting flexibility.

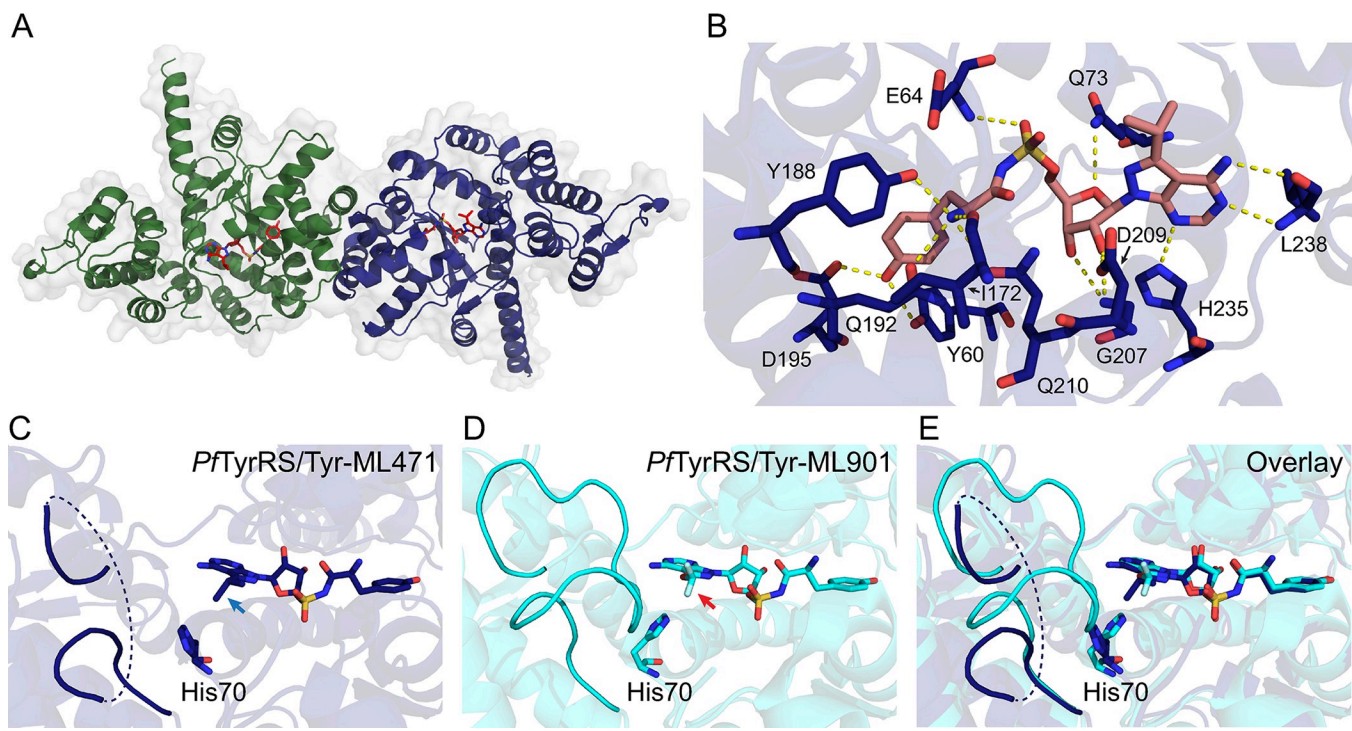

**Fig 6. Comparison of the crystal structures of Tyr-ML471- and Tyr-ML901-bound *Pf*TyrRS reveals differential mobility of the "KMSKS" loop.** (A) Crystal structure of the dimeric *Pf*TyrRS/Tyr-ML471 complex showing chain A (green), chain B (blue), and bound Tyr-ML471 (red, stick representation). (B) Architecture of the B-chain of *Pf*TyrRS with bound Tyr-ML471, showing direct interactions with active site residues. (C) B-chain of Tyr-ML471-bound *Pf*TyrRS showing the poses adopted by the ML471 isopropyl group (blue arrow) and His70, which are incompatible with a structured KMSKS loop. (D) B-chain of Tyr-ML901-bound *Pf*TyrRS (7ROS). The conformation of the ML901 difluoromethoxy group (red arrow) allows His70 to interact with Met248 of the KMSKS loop, leading to stabilisation. (E) Overlay of the B-chains of Tyr-ML471- and Tyr-ML901-bound *Pf*TyrRS.

Comparison of the B chains of the Tyr-ML901-bound and Tyr-ML471-bound *Pf*TyrRS structures highlights a difference in the KMSKS loop organisation. The difluoromethoxy group of ML901 is oriented away from His70 (Fig 6D, red arrow), allowing His70 to adopt a configuration that makes close contact with Met248 in the $_{247}$KMSKS$_{251}$ loop, thereby stabilising the loop (Fig 6D and 6E; aqua backbone). By contrast the isopropyl group of ML471 in chain B is oriented with one methyl group towards His70 (Fig 6C, blue arrow), and the His70 side chain adopts a rotamer that is incompatible with the Met248 interaction observed for Tyr-ML901. Thus, for chain B, Tyr-ML471 binding appears to be associated with loop destabilisation while Tyr-ML901 binding is associated with loop stabilisation. Interestingly, in the A-chain, His70 adopts a similar pose that does not support interaction with Met248 in both Tyr-ML471- and Tyr-ML901-bound structures, leading to destabilisation of the loop (S9C-S9E Fig).

Movement of the KMSKS loop is required for access to the active site. For example, in tRNA$^{Trp}$-bound human TrpRS, the equivalent KMSAS loop adopts a semi-open conformation, that is intermediate between the open conformation observed in the unliganded enzyme and the closed conformation observed in the Trp-AMP complex [37]. The pose adopted by the 7-position isopropyl group of ML471 appears to re-position His70, leading to increased flexibility of the KMSKS loop. This may, in turn, enhance the binding or re-binding of the Tyr-tRNA product, positioning the Tyr-tRNA carbonyl carbon for attack by the sulfamate nitrogen of ML471. This may underpin the higher potency of ML471 as a hijacking inhibitor of recombinant *Pf*TyrRS and as an inhibitor of the growth of *P. falciparum*.

In conclusion, we have identified ML471 as a pyrazolopyrimidine sulfamate with improved potency and selectivity compared with ML901. The improved potency derives from improved efficiency of reaction hijacking inhibition of *Pf*TyrRS and is consistent with enhanced thermal stability induced by the Tyr-ML471 adduct. The enhanced potency of ML471 is associated with repositioning of His70 in the active site of chain B when Tyr-ML471 is bound. The enhanced cellular selectivity of ML471 may be due to its decreased activity against human UAE, which is associated with a lack of interaction between the 7-position substituent of ML471 and active site residues. ML471 exhibits a rapid mode of action and a long *in vivo* half-life underpinning its single dose efficacy in a mouse model of *P. falciparum* malaria. With further improvement of the oral bioavailability, ML471 represents a very interesting compound for prophylaxis, treatment and blocking of transmission of deadly malaria infections.

## Materials and methods

### Ethics statement

Human biological samples were sourced ethically; and their research use was in accord with the terms of the informed written consent, which was obtained from all the participants. Animal studies were ethically reviewed by the GSK Institutional Animal Care and Use Committee, by the Takeda Institutional Animal Care and Use Committee, or the Swiss Tropical and Public Health Institute Animal Ethics Committee, and carried out in accordance with relevant countries' directives, including the European Directive 2010/63/EU, and the relevant Institution's and GSK's Policy on Care, Welfare and Treatment of Animals. Parasitology work and volunteer human blood donation (from healthy adult consenting volunteers) at the University of Pretoria is covered under ethical approval from the Research Ethics Committee from Health Sciences (506/2018) and Natural and Agricultural Sciences (180000094). Studies of *P. vivax* isolates and *P. falciparum* Brazilian isolates were approved by the Ethics Committee of the Tropical Medicine Research Center (Centro de Pesquisa em Medicina Tropical)—CEPEM (CAAE 61442416.7.0000.0011).

### Inhibition of growth of *P. falciparum* cultures

Routine analyses of antimalarial activity against *P. falciparum* 3D7 was tested by TCGLS (Kolkata, India) using the lactate dehydrogenase (*Pf*LDH) growth inhibition assay as previously described [38]. For assays readout, 70 µL of freshly prepared reaction mix containing 100 mM Tris-HCl pH 8, 143 mM sodium L-lactate, 143 µM 3-acetyl pyridine adenine dinucleotide (APAD), 179 µM Nitro Blue tetrazolium chloride (NBT), diaphorase (2.8U/mL) and 0.7% Tween 20 was added into each well of the assay plate. Plates were shaken to ensure mixing and were placed in the dark at 21°C for 20 min. Data were normalized to the percentage of growth inhibition with respect to positive (0.2% DMSO, 0% inhibition) and negative (mixture of 100 µM chloroquine and 100 µM atovaquone, 100% inhibition) controls. *P. falciparum* strain (3D7) was obtained from BEI Resources.

### Analysis of inhibition of growth and viability of *P. falciparum* cultures followed pulsed compound exposure

Assessment of the killing activity of ML901 in pulsed exposure assays was performed using a modification of a previously described procedure [39]. Briefly, cultures of Cam3.II[rev] [40] trophozoites (1.5% final hematocrit; 1.4% final parasitemia) were pre-synchronized to a 5-h window at trophozoite or schizont stage [41]. Compounds were serially diluted in complete medium in v-bottomed microplates. Parasites were exposed to ML901 for 3 h, 6 h, 9 h or 24 h

before washing 5 times with 200 µl of complete medium, then returned to culture conditions. For the no wash samples, ML901 was left in the culture until the assay point. Growth inhibition was assessed in the second cycle by labelling with the DNA-binding dye, SYBR Green I. Quantification of total DNA level reflects cytostatic effects as well as cytocidal effects [42]. Old media (140 µl) was firstly replaced with fresh media, followed by the addition of lysis buffer (20 mM Tris, pH 7.5, 5 mM EDTA, 0.008% w/v saponin, 0.08% v/v Triton X-100) containing SYBR Green I. Plates were incubated at room temperature for 2 h and fluorescence readings were taken using a microplate reader (BMG LABTECH). Unwashed samples containing compounds at >10 times the $IC_{50}$ values were used as background controls.

## Activity against HepG2 cell cultures

The HepG2 (Human Caucasian hepatocyte carcinoma) cell line was procured from ATCC (American Type Culture Collection, Manassas, USA; HB-8065) and viability assessed using CellTiter-Glo luminescent cell viability assay reagents (Promega). For the assay, 2,000 cells/well were plated in 384-well plates 24 h prior to the experiment and incubated at 37˚C in a $CO_2$ incubator. The medium was removed; and cells were treated with fresh medium containing either vehicle (0.5% DMSO) or serially diluted compounds or doxorubicin (1.3 nM to 25 µM) in a final volume of 50 µL/well and further incubated for 72 h at 37˚C in a $CO_2$ incubator. In the positive control wells (100% inhibition), cells were treated with 5 µL of 1% Triton X-100. Following incubation, 25 µL of medium was discarded and 25 µl of CellTiter-Glo reagent was added to each well and the plate was kept on a plate shaker for 15 min at 25˚C with shaking at 300 rpm. Luminescence signals were measured in a PHERAstar FSX reader (BMG LABTECH).

## E1-E2 Transthiolation Assays

An Homogeneous Time-Resolved Fluorescence (HTRF) enzyme assay was employed to evaluate compound activity against ATG7 as previously described [12]. In this assay, a Flag-tagged ATG8 homolog (GABA type A receptor-associated protein; GABARAP) is activated by ATG7 and then transthiolated to a GST-tagged E2 (ATG3). The product of the enzyme reaction, *Flag*-GABARAP-ATG3-*GST*, is quantified by measuring FRET between Europium-Cryptate labelled monoclonal anti-Flag M2 (FLAG M2-Eu cryptate; Revvity, Cat# 61FG2KLB) and goat polyclonal antibody against mouse IgG conjugated to allophycocyanin (APC) (Anti-GST IgG conjugated to SureLight-Allophycocyanin) (Revvity, Cat# AD0059G)). The activation and transthiolation of ubiquitin by UAE, activation and transthiolation of NEDD8 by NAE and activation and transthiolation of SUMO by SAE were all assayed in a similar fashion with appropriately tagged ubiquitin-like proteins and E2 conjugating enzymes as described [9, 25].

## Activity against panels of ex vivo field isolates of *P. falciparum* and *P. vivax*

Compounds were assayed on ten *P. vivax* isolates and seven *P. falciparum* Brazilian isolates collected from mono-infected patients, who had signed a written informed consent form to participate in the study, using previously described methods [43]. The initial parasitemia ranged from 2,100–8,000 parasites/µL for *P. vivax* and 3,500–9,000 parasites/µL for *P. falciparum* isolates. Artesunate and chloroquine were assayed in parallel as standard compounds. The analyses included only the isolates that were incubated for ≥ 40 h with the compounds.

## Parasite Reduction Rate (PRR) estimation

PRR was assessed using a standardized method [27]. *P. falciparum* (strain 3D7A, MRA-151), contributed by David Walliker, was obtained through BEI Resources, NIAID, NIH. Cultures

of parasites (~90% ring stage) were treated with compounds for 120 h, with daily renewal. Samples of parasites were taken at 0, 24, 48, 72, 96 and 120 h. Compounds were washed out and four independent 3-fold serial dilutions were established in 96-well plates, with fresh RBCs and culture medium. After 18 days, and again at 22 days, samples were taken to examine growth using SYBR Green I in an EnVision Multilabel Plate Reader (Perkin Elmer) and analysed using Excel and Grafit 5.0 software. The human biological samples were sourced ethically, and their research use was in accord with the terms of the informed consents under an IRB/EC approved protocol.

## Minimum inoculum of resistance

MIR studies were conducted for ML901 and ML471 using a modified "Gate keeper assay" [34]. The in-house $IC_{50}$ for ML901 was determined to be 2.6 ± 0.05 nM (mean ± SEM; N, n = 2,2), while for ML471 the mean $IC_{50}$ and $IC_{90}$ values were determined as 1.45 and 1.99 nM, respectively. For ML901, the parasites (starting parasite inoculums of $1x10^7$ or $1x10^8$ in triplicate) were subjected to pressure at 3 x $IC_{50}$, with recrudescence on days 12–14. For ML471, parasites were subjected to 10 x $IC_{50}$. The different pressure regimens reflect the standard protocols in the Fidock lab used at the times of analysis. Wells were monitored daily by smear during the first seven days to ensure parasite clearance, during which media was changed daily. Thereafter, cultures were screened three times weekly by flow cytometry and smearing, and the selection maintained a consistent drug pressure over 60 days. In both cases, $IC_{50}$ shifts were observed (ranging from two- to 16-fold). Whole-genome sequencing analysis identified CNVs in chromosome 8 segments, always containing the *Pf*TyrRS locus, consistent with our earlier evidence of this as the target of ML901 [16]. Single nucleotide polymorphism (SNP) filtering was lowered to 0.5 allelic balance (AB) but yielded no point mutations in the core genome.

## Gametocyte killing assays

Gametocytogenesis was induced on a tightly synchronised (>97% rings) asexual parasite culture (Pf3D7-pfs16-CBG99 (kind gift from Pietro Alano), 0.5% parasitemia and 6% hematocrit) with a combination of nutrient starvation and a decrease in hematocrit, as previously described [44]. For immature gametocytes (>90% stage II/III), cultures were exposed to 50 mM N-acetyl glucosamine (NAG) on days 1–4 to eliminate residual asexual parasites and harvested at days 5–6. For mature (>95% stage V) gametocytes, NAG treatment occurred from days 3–7 and harvest at day 13. Stage II/III and V gametocyte cultures (2 % gametocytaemia 1.5 % haematocrit, 150 µL/well) were exposed to compounds and incubated at 37˚C for 48 h under hypoxic conditions [45], after which luciferase activity was determined with a non-lysing D-luciferin substrate (1mM in 0.1M citrate buffer, pH 5.5, 100 µL) and bioluminescence was detected with a 2 s integration time with a GloMax-Multi Detection System with Instinct software.

## Activity against *P. falciparum* male and female gamete formation

Inhibition of the viability of stage V male and female gametocytes was assessed in the *P. falciparum* Dual Gamete Formation Assay (*Pf*DGFA) as described previously [46]. Briefly, mature stage V NF54 strain gametocytes were incubated with test molecules for 48 h in 384-well plates at 37˚C. Gametogenesis was then triggered by cooling the plates to room temperature and addition of xanthurenic acid-containing activating medium (also containing anti-*Pf*s25 antibody (Mab 4B7) conjugated to a Cy3 fluorophore). Twenty minutes after activation, exflagellation was imaged in all wells of the plate using a x4 objective and automated brightfield

microscopy. The plate was then incubated at 20˚C for a further 24 h in the dark. Thereafter, female gamete formation was quantified by automated identification of *Pf*s25-positive cells. Automated counts were transformed to percent inhibition values with reference to positive 100 nM Cabamiquine (DDD498) and negative (DMSO) controls. Data represent the means of multiple independent biological repeats.

## Activity against liver stage *P. falciparum*

Activity against liver stage parasites was performed using a modification of published procedures [47,48]. Briefly, cryopreserved human primary hepatocytes (NF175: H1500.H15B+ Lot No. HC0-6, TebuBio or NF135: F00995-P Lot No. IRZ, BioIVT) were thawed and seeded at 18,000 cells per well in collagen-coated 384-well microtiter plates in medium containing 10% heat inactivated fetal bovine serum (hiFBS). Cells were cultured at 37˚C in 5% $CO_2$. For NF175, medium was replaced by fresh medium containing 10% hiFBS, 5 h post plating. For NF135, medium was replaced by medium containing 0.2% BSA, 24 h post plating. 48 h post plating, salivary glands from *Plasmodium*-NF175 or NF135-infected *Anopheles stephensi* mosquitoes were dissected, added to the hepatocytes (10,000 per well/ NF175; or 12,500 per well/ NF135) and allowed to infect for 3 hours. Sporozoites were then aspirated, and compounds diluted in medium containing 10% hiFBS or 0.2% BSA, were added to the hepatocytes. Medium containing 10% hiFBS or 0.2% BSA and compounds was refreshed daily for four days. Hepatocytes were fixed with ice-cold methanol and samples were blocked with 10% hiFBS in PBS. Schizonts were stained with rabbit anti-HSP70 in 10% hiFBS for 1–2 h followed by incubation with a mixture of secondary goat anti-rabbit AlexaFluor 594 antibody and 4′,6-diamidine-2′-phenylindole dihydrochloride (DAPI) in 10% hiFBS for 30 min. Samples were washed with PBS containing 0.05% Tween 20 between different steps. Cells were imaged on a PicoExpress high content imager, and images were analysed automatically using CellReporterXpress software. Data were analysed by logistic regression using a four-parameter (Hill equation) model and a least-squares method to find the best fit.

## Rat pharmacokinetics (PK) analyses

Sprague-Dawley rats (11 weeks old) were sourced from Hilltop Lab Animals, Inc (Scottdale, Pennsylvania, USA). ML901 was formulated in ethanol: dimethyl acetamide (DMAc): PEG400: $H_2O$ at 1:1:4:4 (v/v) and 10% captisol in 50 mM citrate (pH 3.3) for i.v. (1 mg/kg) and p.o. (10 mg/kg) administration to male Sprague-Dawley rats (n = 3 per route of administration). Blood was collected from a jugular cannula at 0.083, 0.25, 0.5, 1, 2, 4, 8 and 24 h post i.v. dosing, and at the same times (except the 0.083-h sample) following oral administration. A portion of the blood samples were processed into plasma. Samples were precipitated with 0.5% formic acid in methanol and the supernatants were analysed by positive ion electrospray LC-MS for the administered compound. Non-compartmental pharmacokinetic parameters were calculated from individual concentration *vs* time profiles using Phoenix 64 WinNonlin (Version 8.1 Certara, Princeton NJ).

## Permeability analysis

Permeability studies were performed as described previously [49]. In brief, Caco-2 cells were cultured for 21–25 days to differentiate them into enterocyte-like cells. The transepithelial electrical resistance (TEER) was measured to ensure tight junction formation and cells with TEER value more than 250 ohms.$cm^2$ were used in the study. On the day of the transport study, cells were washed with warm HBSS buffer and equilibrated with buffer for 60 min. ML471 was added at a concentration of 5 μM (containing 50 μM Lucifer Yellow) into a 24 Transwell cell

plate (apical 210 μL and basal 1000 μL) and buffer was added in the receiver side. Cells were incubated at 37˚C for 60 min and 120 μL aliquots were taken from the receiver side after 30 and 60 min. Samples were mixed with 100 nM carbutamide in acetonitrile (ACN) containing 0.1% formic acid (internal standard). Samples were centrifuged at 2,056 x g for 10 min and the supernatant was collected and analyzed for quantification of the test article by LC-MS [49].

## *P. falciparum* humanized NOD-scid IL2Rnull mouse model

The model using *P. falciparum* $Pf$3D7$^{0087/N9}$ in NODscidIL2Rγ$^{null}$ mice engrafted with human RBCs was adapted from a previously described procedure [50]. Female NODscidIL2Rγ$^{null}$ mice were purchased from Charles River (Germany). Briefly, two engrafted mice/dosing group (females, 20–22 g) were infected intravenously with $2 \times 10^7$ *P. falciparum* ($Pf$3D7$^{0087/N9}$) on day 0. The antimalarial efficacy was assessed following administration (p.o.) of 100 or 200 mg/kg of compound or of 4 daily doses of 50 mg/kg, initiated on day 3 post-infection. The effect on blood parasitemia was measured by microscopic analysis of Giemsa-stained blood smears (on days 3, 4, 5, 6 and 7 post-infection). Mice were euthanized on day 7. Animal were monitored for symptoms of acute toxicity, including decreased activity, sunken flanks, ataxia and blood in faeces.

## Plasma exposure in the infected mouse model

Compound was administered orally to two mice at 25 mg/kg on days 3, 4, 5, 6 after infection. On day 3, blood samples (20 μL) were obtained at time points up to 24 h after the first administration. Protein was precipitated with acetonitrile and the remaining compound was assessed by LC-MS/MS in the selected reaction monitoring mode using HESI ionization in positive ion mode.

## Preparation of *P. falciparum* TyrRS

*Pf*TyrRS was expressed and purified as previously reported [16]. Briefly, the vectors were transformed into *E. coli* BL21(DE3) cells and induced for 3 h at 37˚C with 0.1–0.5 mM IPTG. Cell pellets were resuspended in the lysis buffer containing 50 mM Tris-HCl, pH 8, 500 mM NaCl, 50 mM imidazole, 1 mM TCEP, 1 mg/mL lysozyme and 1x protease inhibitor cocktail (Roche). Cells were lysed by sonication (Microtip, QSonica) and the lysate was clarified by centrifugation. The supernatant was applied to a HisTrap HP column (GE Healthcare), washed, and eluted using a 0–500 mM imidazole gradient. The eluted His-*Pf*TyrRS was dialyzed overnight with the addition of His-tagged TEV protease (L56V/S135G/S219V triple-mutant). Cleaved His tags and TEV protease were removed by running the dialyzed protein through a HisTrap HP column. *Pf*TyrRS was further purified by gel filtration using a HiLoad 16/600 Superdex 200 column (GE Healthcare).

## *In vitro* transcription/ translation of *Pf*tRNA$^{Tyr}$

A T7 RNA polymerase promoter sequence was added to the 5' end of the DNA sequence of *PftRNA$^{Tyr}$*. This DNA template and its complementary strand were custom-synthesised by Sigma-Aldrich. Two oligonucleotides were annealed at 95˚C for 5 min and the double-stranded DNA template was used for *in vitro* transcription. The transcription reaction was incubated at 37˚C overnight. The reaction mixture consists of template DNA, T7 RNA polymerase and NTP mix as per manufacturer's instructions (HiScribe T7 Quick High Yield RNA Synthesis Kit, NEB). On the following day, the reaction mixture was treated with DNase I at 37˚C for 15 min. tRNA was purified using Phenol: Chloroform: Isoamyl alcohol (25:24:1, v/v),

followed by 1M LiCl precipitation and isopropanol precipitation. Purified *PftRNA*<sup>Tyr</sup> was subjected to NAP-25 desalting columns (Cytiva) to remove floating NTPs. The obtained solutions were concentrated with isopropanol precipitation and the *PftRNA*<sup>Tyr</sup> pellets were dissolved in DEPC-treated water.

## ATP consumption assays

Consumption of ATP was measured using a luciferase-based assay as per the manufacturer's instructions (Kinase-Glo Luminescent Kinase Assay, Promega). Reactions were conducted in 50 mM Tris-HCl pH 7.6, 50 mM KCl, 25 mM $MgCl_2$, 0.1 mg/mL BSA, 1 mM DTT, with 200 µM L-tyrosine, 48 µM *Pf*RNA<sup>Tyr</sup>, 10 µM ATP, 25 nM *Pf*TyrRS and 1 unit/mL inorganic pyrophosphatase (yeast) in the presence or absence of 0–200 µM of inhibitors. Reactions were incubated at 37˚C for 1 h.

## Identification of amino acid-ML471 conjugates generated by ML471-treated *Pf*TyrRS and *P. falciparum* cell culture

*In vitro* recombinant *Pf*TyrRS reactions were set up with 2 µM enzyme, 20 µM L-tyrosine, 10 µM ATP, 10 µM ML471 and 4 µM *Pf*tRNA<sup>Tyr</sup>. The reaction buffer consisted of 25 mM Tris, pH 8, 150 mM NaCl, 5 mM $MgCl_2$ and 1 mM TCEP. The mixture was incubated at 37˚C for 1 h. An equal volume of 8 M urea was added to the mixture after the incubation, followed by trifluoroacetic acid to a final level of 1%. The sample was centrifuged at 15,000 g for 10 min and the supernatant was used for mass spectrometry analysis.

For the identification of adducts in parasite culture, aliquots of late trophozoite stage *P. falciparum* (3D7 strain) culture were exposed to 1 µM ML471 for 2 h. Following treatment, parasite-infected RBCs were lysed with 0.1% saponin and the pellet was washed 3 times with ice-cold PBS. *P. falciparum* cell pellets were resuspended in one volume of water, followed by the addition of five volumes of cold chloroform-methanol (2:1 [vol/vol]) solution. Samples were incubated on ice for 5 min, subjected to vortex mixing for 1 min and centrifuged at 13,500 x g for 10 min at 4˚C to form 2 phases. The top aqueous layer was transferred to a new tube and subjected to LCMS analysis.

High-performance liquid chromatography (HPLC) and mass spectrometric (MS) analyses Samples were analysed by reversed-phase ultra-high performance liquid chromatography (UHPLC) coupled to tandem mass spectrometry (MS/MS) (Q Exactive, ThermoFisher Scientific). Samples (5 µL) were injected onto a Dionex Ultimate 3000 UHPLC system (ThermoFisher Scientific) and analytical separation was performed with a RRHD Eclipse Plus C8 column (2.1 × 100 mm, 1.8 µm; Agilent Technologies). The system was run at a flow rate of 300 µL/min using a binary gradient solvent system consisting of 0.1% formic acid in water (solvent A) and 0.1% formic acid in acetonitrile (solvent B). The gradient profile was as follows: 0–4.5 min, 3–40% B; 4.5–4.6 min, 40–95% B; 4.6–5.5 min, 95% B; 5.5–5.8 min, 95–3% B and 5.8–8 min, 3% B. Full scan MS acquisition was performed in polarity switching mode, with the following settings: resolution 35,000, 900 AGC target $1 \times 10^6$, *m/z* range 85–1275, sheath gas 50, auxiliary gas 20, sweep gas 2, probe temperature 120˚C, capillary temperature 300˚C and S-Lens RF level was set to 50. The spray voltage was set at 3.5 kV for positive and negative ionization modes. All data shown for the Tyr-ML471 adduct were collected using positive mode ionisation.

## Differential scanning fluorimetry (DSF)

The effect on ML471 and analogues on the thermal stability of TyrRS enzymes was assayed as previously described [16]. Briefly, *Pf*TyrRS (2.3 µM) was incubated with 50 µM nucleoside

sulfamate (molar ratio 1: 22) with 10 μM ATP, 20 μM L-tyrosine, 4 μM *Pf*tRNA$^{Tyr}$ and at 37˚C for 2 h. Alternatively, *Hs*TyrRS (2.3 μM) was incubated with 200 μM nucleoside sulfamate (molar ratio 1: 87) with 10 μM ATP, 20 μM L-tyrosine, 8 mg/mL yeast tRNA, at 37˚C for 4 h. A longer incubation and higher nucleoside sulfamate concentration was used for *Hs*TyrRS, given its weaker susceptibility to hijacking. SYPRO Orange (Sigma-Aldrich; 5,000X concentrate in DMSO) was added to the reaction mixture at a final concentration of 5X. 25 μL of the sample was added into each well of a 96-well qPCR plate (Applied Biosystems). The plate was sealed and analysed using StepOnePlus Real-Time PCR system (Applied Biosystems). The samples were heated from 20˚C to 90˚C with a 1% continuous gradient. The thermal unfolding curve was plotted as the first derivative curve of the raw fluorescence values. The melting temperature ($T_m$), defined as the peak of the first derivative curve, was used to assess the thermal stability of protein-ligand complexes.

## Crystallography

Recombinant *Pf*TyrRS in complex with synthetic Tyr-ML471 was crystallised using the sitting drop vapour diffusion technique at 20˚C. Crystals were formed in 2.25 M sodium malonate, pH 6. Drops contained 1.5 μL of protein-ligand solution (10 mg/mL *Pf*TyrRS in Tris-HCl (25 mM, pH 8)), 100 mM NaCl, 10 mM MgCl$_2$, 1 mM TCEP, 500 μM Tyr-ML471 synthetic ligand) and 1.5 μL of crystallant solution (2.25 M sodium malonate, pH 6).

Crystals were flash-cooled in liquid nitrogen directly from the crystallisation drop, and X-ray diffraction data were collected at the MX2 beamline of the Australian Synchrotron [51]. Diffraction data were indexed and integrated using XDS and analysed using POINTLESS, prior to merging by AIMLESS [52] from the CCP4 software suite [53]. Initial phase estimates for *Pf*TyrRS in complex with Tyr-ML471 were obtained by molecular replacement in PHASER [54] using modified crystal structure coordinates of *Pf*TyrRS/ Tyr-ML901 as the search model (PDB ID: 7ROS, Xie 2022). Automated structure refinement using phenix.refine [55] was followed iteratively by manual model building in COOT [56]. Structure refinement was performed using non-crystallographic torsion restraints and translation/libration screw (TLS) refinement with each chain comprising a single TLS group. Restraints for Tyr-ML471 were generated using phenix.elbow [57]. Final data collection and refinement statistics are shown in S13 Table.

## Docking

Docking was carried out with the Surflex-Dock molecular docking module in SybylX2.1 (Certara, NJ, USA). Docks were performed both with and without protein flexibility. Docking poses with the best Surflex scores were inspected in SybylX2.1 and figures generated with PyMOL.

## Chemistry

The syntheses of the compounds from the series have been reported previously [12, 22]. MMV designations for the compounds are listed in Table 1. Methods used for resynthesis of ML471; and for synthesis of Tyr-ML471 are presented in S1 Text. Adenosine 5'-sulfamate (AMS) [58] and was kindly provided by Dr Derek Tan, Memorial Sloan Kettering Cancer Center.

## Supporting information

**S1 Fig. Exposure-time dependent responses of trophozoite and schizont stage parasites to short pulses of ML901.** (A,B) A tightly synchronized culture of CAM 3.II Rev parasites

(>70% of parasites within a 5-h time window) was subjected to pulses of ML901 for 3 h, 6 h, 9 h, 24 h, or continued exposure for 48 h (A) or 30 h (B), initiated at (A) trophozoite (25–30 h.p. i.) and (B) schizont (43–48 h.p.i.) stages. Flow cytometric analysis of Syto-61-labelled parasites in the cycle after the initiation of treatment assessed cell viability. Data are representative of three (trophozoite) and two (schizont) independent experiments. $LD_{50}$ values are shown in Table S1.
(TIF)

**S2 Fig. Activity of ML471, artesunate and chloroquine against *ex vivo* field isolates.** Compounds were assayed on *P falciparum* (A) isolates and *P. vivax* (B) Brazilian isolates collected from mono-infected patients. Data represent mean ± SEM. Median $EC_{50}$ (nM) values, the range of values, and the numbers of isolates are shown in Table S2.
(TIF)

**S3 Fig. Activity of ML471, ML901, MMV390048 and Methylene blue (MB) against immature (A) and mature (B) stage gametocytes.** Gametocytocidal activity of the compounds was assessed against immature (>90% stage II/III) and mature (>95% stage V) stage gametocytes. MMV390048 and MB were used as controls. Data represent mean ± SEM from three independent experiments. Mean $IC_{50}$ ± SEM values from the three independent experiments are presented in S3 Table.
(TIF)

**S4 Fig. Activity of ML471 and ML901 against *P. falciparum* NF175 (A) and NF135 (B) liver stage schizonts.** Human primary hepatocytes were infected with *P. falciparum* NF175 or NF135 sporozoites and cultured for four days. Anti-HSP70 was used to detect parasites in fixed cells using high content imaging. Atovaquone and MMV390048 were used as control compounds against NF175 and NF135 schizonts, respectively. Data values represent mean ± SEM from three independent experiments. $IC_{50}$ values are shown in S4 Table.
(TIF)

**S5 Fig. Activity of ML471 and Cabamiquine (DDD498) against transmissible gametes.** Inhibition of male (A) and female (B) gamete formation was assessed in the *P. falciparum* Dual Gamete Formation Assay. Cabamiquine (DDD498) and DMSO were used as positive and vehicle controls, respectively. Data represent mean ± SEM from 4–5 independent experiments. $IC_{50}$ values are shown in S5 Table.
(TIF)

**S6 Fig. Pharmacokinetics profile and efficacy of ML471 in treating *P. falciparum* infected mice.** (A) Pharmacokinetics profile (in blood) for SCID mice engrafted with human RBCs infected with *P. falciparum*, over the first day following treatment with ML471 at 50 mg/kg p. o. See S8 Table for pharmacokinetics values. (B) Therapeutic efficacy of ML471 in the SCID mouse *P. falciparum* model, dosed with ML471 for 4 days at 50 mg/kg p.o. per day (arrows), initiated on Day 3 post-infection. The chloroquine data are from [16].
(TIF)

**S7 Fig. Identification of Tyr-ML471 adduct made by *P. falciparum* culture and *Pf*TyrRS enzyme.** (A) MS/MS analysis of the Tyr-ML471 adduct made by *P. falciparum* following treatment with ML471 (1 μM) for 2 h (upper panel); and the synthetic conjugate at 0.2 μM (lower panel). (B,C) *Pf*TyrRS (1 μM) was incubated with ML471 (10 μM), ATP (10 μM), tyrosine (20 μM) and 4 μM *Pf*tRNA$^{Tyr}$ for 1 h at 37°C. Following protein denaturation and precipitation, the supernatant was subjected to LCMS analysis. (B) The extracted ion chromatograms of Tyr-ML471 adduct made by *Pf*TyrRS (upper panel); and the synthetic conjugate at 1 μM

(lower panel). The inset shows the MS analysis of the enzyme-generated Tyr-ML471. (C) MS/MS analysis of the enzyme-generated Tyr-ML471 (upper panel) and the synthetic conjugate at 1 μM (lower panel).

(TIF)

**S8 Fig. Kinase GLO assays for ML901 and derivatives.** Effects of increasing concentrations of ML723, ML111, ML864, ML470 (A) and ML107, ML676, ML681 (B) on ATP consumption by *Pf*TyrRS. The reaction conditions are: *Pf*TyrRS (25 nM), ATP (10 μM), tyrosine (200 μM), cognate tRNA$^{Tyr}$ (4.8 μM) and pyrophosphatase (1 unit/mL). Incubations were at 37°C for 1 h. Data are mean ± SEM from three independent experiments.

(TIF)

**S9 Fig. Comparison of the active site architecture of *Pf*TyrRS in complex with Tyr-ML471 or Tyr-ML901 (7ROS).** (A) Inhibitor/active site interactions for the A-chain of *Pf*TyrRS with bound Tyr-ML471. (B) LigPlots of interacting residues for the A- and B-chains of *Pf*TyrRS with bound Tyr-ML471. (C) A-chain of Tyr-ML471-bound *Pf*TyrRS showing the poses adopted by the ML471 isopropyl (aqua arrow) and His70, which are incompatible with a structured KMSKS loop. (D) A-chain of Tyr-ML901-bound *Pf*TyrRS (7ROS) illustrating the ML901 difluoromethoxy group (red arrow) and the His70 conformation. The KMSKS loop is not resolved. (E) Overlay of the A-chains of Tyr-ML471- and Tyr-ML901-bound *Pf*TyrRS.

(TIF)

**S1 Table. The median lethal dose (LD$_{50}$) values of different asexual blood stages after exposure of ML901.**

(PDF)

**S2 Table. *P. vivax* and *P. falciparum* ex vivo drug susceptibility.**

(PDF)

**S3 Table. Activity against immature (>90% stage II/III) and mature (>95% stage V) stage gametocytes (*Pf*3D7-pfs16-CBG99) for ML901, ML471 and antimalarial controls.**

(PDF)

**S4 Table. Activity against liver stage *P. falciparum* schizonts and against primary hepatocytes.**

(PDF)

**S5 Table. Activity against transmissible gametes.**

(PDF)

**S6 Table. Parasite Reduction Ratio.**

(PDF)

**S7 Table. Pharmacological and ADME characterization of ML471.**

(PDF)

**S8 Table. Pharmacological parameters for ML471 in the SCID mouse model.**

(PDF)

**S9 Table. Summary of resistance selection.**

(PDF)

**S10 Table. Copy Number Variants (CNV) in ML901-selected sample.**

(PDF)

**S11 Table. Copy Number Variants (CNV) in ML471-selected sample.**
(PDF)

**S12 Table. $T_m$ values of TyrRSs determined by differential scanning fluorimetry (DSF).**
(PDF)

**S13 Table. X-ray diffraction data collection and refinement statistics.**
(PDF)

**S1 Text. Chemistry materials and methods.**
(DOCX)

## Acknowledgments

We thank the following colleagues for technical contributions: Liver Stage Assay: Marloes de Bruijni and Rob Henderson, TropIQ, Netherlands; Caco2 Assays: Bei-Ching Chuang, Takeda Pharmaceuticals, USA; SCID mouse assay: Ursula Lehmann, Swiss Tropical and Public Health Institute, Switzerland, Christoph Siethoff, Swiss BioQuant; 3D7 parasite assays: TCG Life-Sciences, Kolkata, India; Assay coordination: Delphine Baud and Anna Adam, Medicines for Malaria Venture, Switzerland; Mass Spectrometry: Shuai Nie and Nick Williamson, Melbourne Mass Spectrometry and Proteomics Facility; Crystallisation: Roxanne Smith, Bio21-WEHI Crystallisation facility; Protein Purification: Yee-Foong Mok, Melbourne Protein Facility, Bio21 Institute. We thank Hirotake Mizutani, Takeda Pharmaceuticals and Winnie Ye, University of Melbourne, for technical help. Thanks also extended to Heekuk Park, Sachel Mok and Anne-Catrin Uhlemann for whole-genome sequencing and analysis at the Columbia University Irving Medical Center. We thank Dr Derek Tan, Memorial Sloan Kettering Cancer Center, for supplying adenosine 5'-sulfamate. We thank the Australian Red Cross for supply of blood products. GSK acknowledges the Centro de Hemoterapia y Donación de Valladolid, Castilla y León, and the Centro de Transfusiones de la Comunidad de Madrid for the supply of blood samples. This research was partly undertaken at the Australian Synchrotron, part of the Australian Nuclear Science and Technology Organization, and made use of the ACRF Detector on the MX2 beamline. We thank the beamline staff for their assistance. The University of Pretoria Institute for Sustainable Malaria Control (LMB) acknowledges the South African Medical Research Council as Collaborating Centre for Malaria Research.

## Author Contributions

**Conceptualization:** Stanley C. Xie, Chia-Wei Tai, Craig J. Morton, Liting Ma, Sergio Wittlin, Con Dogovski, Francisco J. Gamo, Craig A. Hutton, David A. Fidock, Lawrence R. Dick, Stephen L. Brand, Alexandra E. Gould, Steven Langston, Michael D. W. Griffin, Leann Tilley.

**Formal analysis:** Stanley C. Xie, Chia-Wei Tai, Craig J. Morton, Liting Ma, Shih-Chung Huang, Sergio Wittlin, Yawei Du, Yongbo Hu, Con Dogovski, Mina Salimimarand, Robert Griffin, Dylan England, Elisa de la Cruz, Ioanna Deni, Tomas Yeo, Anna Y. Burkhard, Josefine Striepen, Kyra A. Schindler, Benigno Crespo, Yogesh Khandokar, Tayla Rabie, Lyn-Marié Birkholtz, Mufuliat T. Famodimu, Michael J. Delves, Judith Bolsher, Karin M. J. Koolen, Rianne van der Laak, Anna C. C. Aguiar, Dhelio B. Pereira, Rafael V. C. Guido, David A. Fidock, Lawrence R. Dick, Stephen L. Brand, Alexandra E. Gould, Steven Langston, Michael D. W. Griffin, Leann Tilley.

**Funding acquisition:** Stanley C. Xie, Sergio Wittlin, Francisco J. Gamo, Craig A. Hutton, Lyn-Marié Birkholtz, Rafael V. C. Guido, Darren J. Creek, David A. Fidock, Alexandra E. Gould, Steven Langston, Michael D. W. Griffin, Leann Tilley.

**Investigation:** Stanley C. Xie, Chia-Wei Tai, Craig J. Morton, Liting Ma, Shih-Chung Huang, Sergio Wittlin, Yawei Du, Yongbo Hu, Con Dogovski, Mina Salimimarand, Robert Griffin, Dylan England, Elisa de la Cruz, Ioanna Deni, Tomas Yeo, Anna Y. Burkhard, Josefine Striepen, Kyra A. Schindler, Benigno Crespo, Yogesh Khandokar, Tayla Rabie, Mufuliat T. Famodimu, Michael J. Delves, Judith Bolsher, Karin M. J. Koolen, Rianne van der Laak, Anna C. C. Aguiar, Dhelio B. Pereira.

**Writing – original draft:** Stanley C. Xie, Chia-Wei Tai, Craig J. Morton, Liting Ma, Sergio Wittlin, Mina Salimimarand, Craig A. Hutton, David A. Fidock, Lawrence R. Dick, Stephen L. Brand, Alexandra E. Gould, Steven Langston, Michael D. W. Griffin, Leann Tilley.

**Writing – review & editing:** Stanley C. Xie, Chia-Wei Tai, Craig J. Morton, Liting Ma, Shih-Chung Huang, Sergio Wittlin, Yawei Du, Yongbo Hu, Con Dogovski, Mina Salimimarand, Robert Griffin, Dylan England, Elisa de la Cruz, Ioanna Deni, Tomas Yeo, Anna Y. Burkhard, Josefine Striepen, Kyra A. Schindler, Benigno Crespo, Francisco J. Gamo, Yogesh Khandokar, Craig A. Hutton, Tayla Rabie, Lyn-Marié Birkholtz, Mufuliat T. Famodimu, Michael J. Delves, Judith Bolsher, Karin M. J. Koolen, Rianne van der Laak, Anna C. C. Aguiar, Dhelio B. Pereira, Rafael V. C. Guido, Darren J. Creek, David A. Fidock, Lawrence R. Dick, Stephen L. Brand, Alexandra E. Gould, Steven Langston, Michael D. W. Griffin, Leann Tilley.

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
