## [Decision Letter · Decision Letter 0]

19 Sep 2024

Dear Prof. Tilley,

Thank you very much for submitting your manuscript "A potent and selective reaction hijacking inhibitor of Plasmodium falciparum tyrosine tRNA synthetase exhibits single dose oral efficacy in vivo" for consideration at PLOS Pathogens. As with all papers reviewed by the journal, your manuscript was reviewed by members of the editorial board and by several independent reviewers. The reviewers appreciated the attention to an important topic. Based on the reviews, we are likely to accept this manuscript for publication, providing that you modify the manuscript according to the review recommendations.

Sincerely,

Audrey Ragan Odom John, MD, PhD

Academic Editor

PLOS Pathogens

Tracey Lamb

Section Editor

PLOS Pathogens

Michael Malim

Editor-in-Chief

PLOS Pathogens

orcid.org/0000-0002-7699-2064

Reviewer Comments (if any, and for reference):

Reviewer's Responses to Questions

**Part I - Summary**

Reviewer #1: In the manuscript titled “A potent and selective reaction hijacking inhibitor of Plasmodium falciparum tyrosine tRNA synthetase exhibits single dose oral efficacy in vivo,” Xie et al explore ML901 derivatives to identify a candidate with improved selectivity for future antimalarial development. This led to the discovery of ML471, which exhibits activity against the P. falciparum asexual blood stage, sexual stage and liver stage. Activity against clinical isolates (falciparum and vivax) as well as drug resistant P. falciparum was confirmed. The possibility that ML901 exhibits cytotoxicity due to human E1 targeting was explored and ML471 was found to have decreased targeting of human UAE. ML471 was tested in an in vivo mouse malaria model where it had single dose oral efficacy against P. falciparum and rapid killing, meeting current MMV recommendations for new candidate selection criteria. Though it is acknowledged that it does not meet goals for reduced propensity of resistance. Lastly, structural studies were completed that provide insights into the mechanism of selectivity. The experiments presented are robust, and the authors nicely frame their findings in the context of previously published work and the current state of targeting tRNA synthetases. This impactful and timely manuscript will be of broad interest to the pathogen and drug development communities.

Reviewer #2: The manuscript is very well written and serves to be a very good example of the strength of collaborative work. The authors have indeed done a commendable amount of work and have very clearly justified every aspect of the compound testing that they performed. All of their claims are very well justified with the experimental results they have. The significance of this work can be observed by virtue of the establishment of a whole “pipeline” to effectively scrutinize multiple aspects of a compound in drugging the malarial parasite. This work, with its immense width-and-depth analyses positions itself to form a basis of future works utilizing the concept of reaction-hijacking prodrugs in more use-cases. The novelty of chemical classes being discovered and synthesized to have desirable bioactive profiles promises to continue to be an evergreen reiterative process to push the limits of molecular medicine.

**Part II – Major Issues: Key Experiments Required for Acceptance**

Reviewer #1: Figure 4A, could a control (no drug) trace be added to demonstrate no peak is observed in parasites without compound addition.

- The magnitude of a Tm shift (driven by thermodynamics) is not proportional with ligand binding affinity (pg 10, line 248). Perhaps reword this to discuss a more stable conformational change induced by the small molecule. Alternatively, ITC or MST could be used to measure the Kd. It is also recommended to remove this suggestion (that enhanced thermal stability indicated tighter binding) in line 337.

Reviewer #2: No major issues detected in the manuscript.

**Part III – Minor Issues: Editorial and Data Presentation Modifications**

Reviewer #1: - Is there a structural rationale for the 8 analogs tested?

-Can the authors speculate on the loss of efficacy observed between the sexual and asexual stage? Also, has the activity of ML471 or ML901 on parasite invasion been investigated?

- pg 7, line 107 could the notation be added to indicate it was at 72 hr for the IC50.

- On pg 9, line 210 it is stated that dosing led to “no evidence of toxicity.” Could the assessment for toxicity be clarified?

-Could Table S9 include previously published metrics for compounds that have favorable Minimum Inoculum for Resistance values for comparison?

- These compounds bind/modify the protein in the nM range but inhibit ATP consumption in the uM range. Could a rational for this be included?

- Conventionally prodrugs (or proinhibitors) require modification by cytochrome P450s, or other enzymes, to yield active molecules. Here, it seems pro-inhibitor is indicating the molecule pre modification of the target? To the best of my knowledge, this terminology has not been used for covalent kinase inhibitors.

Reviewer #2: Line 218 – For the in vitro resistance evolution experiment, what was the rationale for the authors to use two differential methods of inducing resistance?

Line 587 – Please mention the ratio of “Enzyme: Compound” for both the enzymes tested and the associated rationale of performing the assay at different ratios for the Pf and Hs enzymes. Currently only the PfTyrRS parameters are mentioned.

Line 761 – Please provide a full legend for the Table contents as to what is each column value depicting.

Line 766 – Please provide a full legend for the Table contents as to what is each column value depicting.

Line 772 – Please provide a full legend for the Table contents as to what is each column value depicting. Please also move the description of the assay parameters to the DSF section in the methods. Possible to add a table in the supplementary tables mentioning the raw melt temperatures by virtue of which the delta Tms have been calculated?

Please add all the relevant graphs in the supplementary figures for the supplementary tables S1, S2, S3, S4 and S5.

Please refer to every graph, figure or table wherever a related point is referred to in the text. In some cases, it might get difficult for a reader to connect two points which might be obvious to specialists in the field.

PLOS authors have the option to publish the peer review history of their article (what does this mean?). If published, this will include your full peer review and any attached files.

Reviewer #1: No

Reviewer #2: No

Figure Files:

Data Requirements:

Reproducibility:

References:

---

## [Editor Report · Decision Letter 1]

31 Oct 2024

Dear Prof. Tilley,

We are pleased to inform you that your manuscript 'A potent and selective reaction hijacking inhibitor of Plasmodium falciparum tyrosine tRNA synthetase exhibits single dose oral efficacy in vivo' has been provisionally accepted for publication in PLOS Pathogens.

Best regards,

Audrey Ragan John, MD, PHD

Academic Editor

PLOS Pathogens

Tracey Lamb

Section Editor

PLOS Pathogens

Michael Malim

Editor-in-Chief

PLOS Pathogens

orcid.org/0000-0002-7699-2064
---

## [Editor Report · Acceptance letter]

27 Nov 2024

Dear Professor Tilley,

We are delighted to inform you that your manuscript, "A potent and selective reaction hijacking inhibitor of *Plasmodium falciparum* tyrosine tRNA synthetase exhibits single dose oral efficacy *in vivo*," has been formally accepted for publication in PLOS Pathogens.

Best regards,

Michael Malim

Editor-in-Chief

PLOS Pathogens

orcid.org/0000-0002-7699-2064